# MARVIS: Modality Adaptive Reasoning over VISualizations

## Abstract

Predictive applications of machine learning often rely on small (sub 1 Bn parameter) specialized models tuned to particular domains or modalities. Such models often achieve excellent performance, but lack flexibility. LLMs and VLMs offer versatility, but typically underperform specialized predictors, especially on non-traditional modalities and long-tail domains, and introduce risks of data exposure. We propose MARVIS (Modality Adaptive Reasoning over VISualizations), a training-free method that enables small vision-language models to solve predictive tasks on any data modality with high accuracy, and without exposing private data to the VLM. MARVIS transforms latent embedding spaces into visual representations and then leverages the spatial and fine-grained reasoning skills of VLMs to interpret the visualizations and utilize them for predictions successfully. MARVIS achieves competitive performance across vision, audio, biological, and tabular domains using a single 3B parameter model, yielding results that beat Gemini 2.0 by 16% on average. MARVIS drastically reduces the gap between LLM/VLMs approaches and specialized domain-specific methods, without exposing sensitive data or requiring any domain-specific training. We open source our code and datasets at https://anonymous.4open.science/r/marvis-6F54

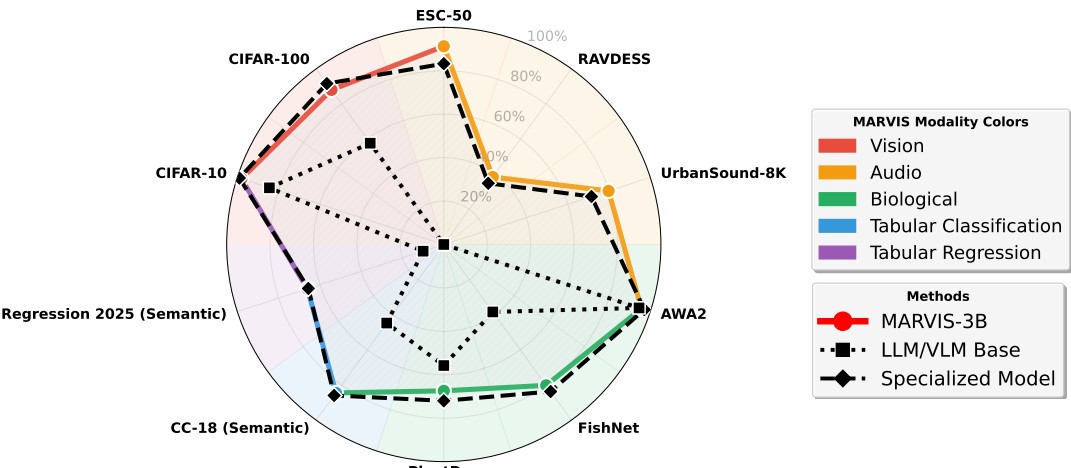

Figure 1: **MARVIS transforms VLMs into frontier predictors.** Using a standard 3B parameter QwenVL model zero-shot without reasoning, MARVIS (colored line) achieves competitive performance compared to specialized baselines (dashed line) across modalities and domains, far exceeding the best existing LLM / VLM predictors (dotted line).

# 1  INTRODUCTION

Much of the progress in the field of machine learning in recent years has been on classification and regression tasks (which, in this work, we sometimes collectively refer to as *predictive* tasks). These have historically been addressed either using classical machine learning methods or, more recently, with deep learning. In the latter case, the best performance has generally been achieved using **specialized models** with less than one billion parameters tuned for a particular task and/or knowledge domain (Prokhorenkova et al., 2018; He et al., 2015; Hollmann et al., 2025). These models often learn to compress a high-dimensional input space into a simplified embedded space; these embeddings can then be used for prediction without any fine-tuned classification stage via classical nonparametric methods like KNN (Oquab et al., 2023) or parametric fine-tuning. What these models gain in precision, however, they sacrifice in flexibility. Narrow experts are often inapplicable to other domains without additional fine-tuning (Devlin et al., 2019).

**LLM and VLMs** introduced an exciting new paradigm: in-context learning (ICL) over text and images, which allowed these models to adapt to new tasks without weight updates (Brown et al., 2020). Gemini, GPT-4V and LLaVA (Liu et al., 2023a) seek to optimally align language models with specialist embeddings for vision, and in some cases, other modalities as well. Unlike specialists, LLMs are extremely flexible; users can ask almost anything in natural language, and in many cases, receive a reasonable response. However, recent research has demonstrated that even state-of-the-art VLMs from OpenAI and Google consistently underperform as predictors when compared to specialist classifiers, especially on non-traditional modalities and in long-tail domains (Zhang et al., 2024). For some modalities, such as audio, there is no obvious way to natively utilize a traditional LLM / VLM for predictive tasks.

But perhaps the most significant weakness of LLMs and VLMs, especially those which can only be used via API endpoints, is the practical and regulatory threat of sensitive data exposure. API providers frequently train on user data, and the models themselves can be prompted to regurgitate sensitive training data verbatim (Kandpal et al., 2024; Nasr et al., 2023). Even when inference providers offer guarantees that user data will not be included in training corpora, trust or regulatory gaps impede many businesses interested in adopting GenAI. Existing solutions, such as locally hosting LLMs and automatically detecting P.I.I., may sacrifice model quality, require extensive infrastructure, or be limited in scope and precision. These challenges motivate our core research question:

> **⊞ Research Question**
>
> How can we combine the reasoning capabilities of LLMs with the representational power of specialized models without requiring modality-specific fine-tuning or exposing sensitive data?

In this work, we posit that visual reasoning, coupled with specialized low-dimensional embedding models, is a skeleton key that unlocks the power of in-context learning and reasoning for arbitrary data modalities and domains, including data that is sensitive.

> **Contributions**
>
> 1. We propose MARVIS, an efficient, modality-agnostic system for transforming a VLM into a performant predictor. Without access to P.I.I. or direct data leakage, using a QwenVL model with no specialized reasoning training, MARVIS achieves competitive performance across vision, audio, and tabular modalities, and across a wide range of scientific domains, on both classification and regression tasks.

2. We demonstrate empirically that MARVIS does more than simply copy predictions; it reasons over the available information sources, implicitly analyzing and balancing them to improve its own predictive power. It can rationalize its decisions post-hoc and suggest next steps, unlike the specialist models it adapts.

3. We also introduce numerous valuable secondary contributions to facilitate future research in this area, including the first large-scale standardized tabular classification and regression datasets with complete semantic information (see Appendix H), a strong FFT baseline for tabular data (see Appendix D), comprehensive ablations, and a well-documented Github repository.

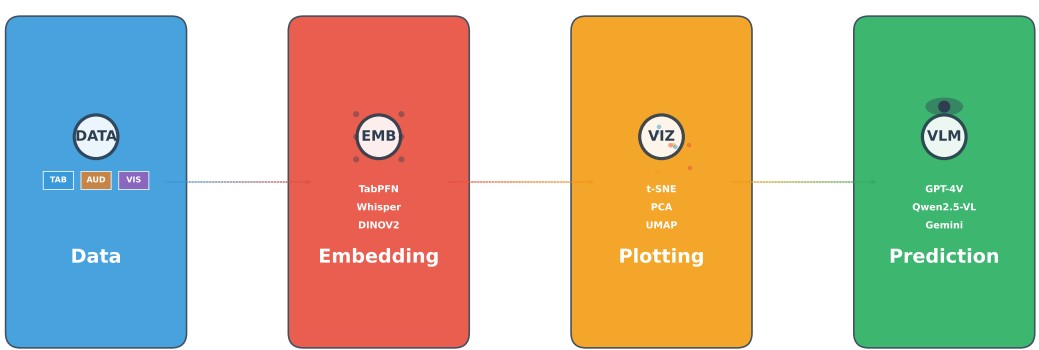

Figure 2: **The four-stage MARVIS pipeline.** We start with raw input data, capture key patterns using specialist embedding generating models, determine an appropriate strategy for plotting the data, and prompt a VLM with visual context, as well as (optionally) metadata and semantic context, then extract predictions.

## 2 MARVIS

**Core Insight: Vision is a Skeleton Key.** Relying solely on text to ingest data is limited and does not align with how humans operate. For predictive tasks, it is not usually the raw data that we want the model to reason over; rather, it is a distilled view of that data, for the purposes of answering specific questions or rendering judgments. Human scholars tend to reason more effectively with data visualizations, simplified views of complex data (Unwin, 2020; de Bodt et al., 2025). VLMs, which are pretrained on web-scraped data, can understand and interpret a wide range of scientific imagery and visualizations of specialized embedding spaces. Thus, we posit that *embedding visualizations are skeleton keys*, enabling us to reason about any kind of data with vision-language models without modality-specific training beyond vision. Moreover, visualizations can be easily generated at inference time with standard packages, such as scikit-learn (Pedregosa et al., 2011).

MARVIS operates through the following pipeline:

1. **Embedding Generation**: Use specialized embedding models to create vector representations.
2. **Dimensionality Reduction**: Apply t-SNE to create 2D visualizations optimized for VLM processing.
3. **Visual Reasoning**: Query the VLM with the visualization and query point for a prediction.
4. **Response Processing**: Extract the prediction from VLM's reasoning.

We present a visual overview of MARVIS in Fig. 2, and in Appendix K, we also provide complete visual examples extracted from our study.

## 2.1 DESIGN CHALLENGES IN VISUAL PREDICTIVE SYSTEMS

Although the principles of MARVIS are extremely simple, in order to apply them in practice, we had to overcome significant technical hurdles.

**Challenges: architecture.** The first is choosing an appropriate VLM architecture; many older architectures either cannot localize what they "see" effectively, or cannot "see" clearly enough to take advantage of visualizations. After some trial and error, we choose the 3B parameter Qwen 2.5 VL model from Alibaba (Bai et al., 2025). This model has several key advantages for our purposes; firstly, it uses 14×14 patches with sliding window attention in some layers, emphasizing local patch interaction. This is important for distance-based visualizations, where proximity matters. Second, it allows images of arbitrary aspect ratios to be processed effectively, without distorting distances during ingestion. This allows us to effectively compose and read multi-visualization layouts with MARVIS. Third, the Qwen 2.5 VL series has been specifically trained to work with long context and scientific imagery. We validate this choice in section E.2, showing that MARVIS-3B matches the performance of GPT-4o-mini and outperforms a much larger recent thinking model from Kimi.

**Challenges: resolution.** Even Qwen 2.5 VL does not "see" as well as humans; the particular patch dimensions and the limited range of its local attention mean that Qwen performs best when DPI is optimized and scaling is utilized to enhance the region of interest. We find that the amount required varies substantially depending on the benchmark, but can usually be set once for each benchmark; this avoids costly hyperparameter search, although this value could conceivably be optimized further in the future. Ideally, the scaling factor is such that the target point and its neighbors are captured within the 14x14 patches from the sliding window, significantly enhancing spatial understanding.

**Challenges: context composition strategy.** One key design decision in MARVIS is which context to include, and how much of it. In Appendix E.1, we name and ablate over 25 different configurations. Ultimately, for our main experiments in this paper, we exclusively use the "tsne_knn" setting, as we find it offers the best speed / quality tradeoff. Because KNN operates on the embeddings without dimensionality reduction, it is sometimes able to discover relationships that visualizations miss; however, we consider this an important area for future research, as we believe we have only begun to document the possibilities here. We find that fixing the nearest neighbors hyperparameter at min(30, 10% of the training data) works well for a wide range of dataset sizes and modalities.

**Challenges: classname extraction.** In order to avoid the common failure mode in which answers are correct but not detected by the parser, we introduce consistent color schemes and consistent naming across the legends for all visualizations, ensuring clear visual separation for VLM interpretation. The parser is made aware of both the class names and the color names, and is given a mapping between them. Classnames in legends are limited to the classes which actually appear in that visualization, in order to control the size of the legend for large datasets.

## 3 EXPERIMENTS

**Overview.** Our main experiments assess MARVIS across four distinct modalities using domain-appropriate embedding models and established benchmarks; we compare against both specialized baselines and alternative LLM/VLM approaches.

Table 1 presents MARVIS performance across all modalities compared to 5 specialized baselines and 4 alternative LLM/VLM approaches. For each benchmark, we conduct a single MARVIS run. We use a QwenVL 2.5 3B Instruct backbone. For each benchmark, we tune T-SNe zoom factor and KNN neighbor

Table 1: **Domain-specific embeddings, benchmarks, and detailed results.** Results are boldfaced when statistically tied for best performance within 95% confidence intervals (normal approximation). MARVIS demonstrates competitive or superior performance on most individual benchmarks, achieving average results within 2.5% of an ensemble of specialized methods while providing universal applicability. Benchmark acronyms: C10 = CIFAR-10, C100 = CIFAR-100, ESC = ESC-50, RAV = RAVDESS, US8 = UrbanSound8K, FSH = FishNet, AWA = AWA2, PLD = PlantDoc, CC18 = OpenML CC18, R25 = Regression 2025. We show the best results of specialized models and traditional LLM/VLM approaches. For all benchmarks except R25, the metric is Accuracy. For R25, it is R2 Score (with a minimum score of 0). The number reported is the mean over all sub-tasks for multi-task benchmarks.

| Domain | Embeddings | Benchmark | Size (K) | MARVIS | Specialized Model | LLM/VLM | 95% CI |
|--------|-----------|-----------|----------|--------|------------------|---------|--------|
| Vision | DINOV2 | C10 | 60 | 98.0 | **99.0** (DINOV2) | 85.7 (Gemini) | ±0.1 |
|        |          | C100 | 60 | 88.0 | **91.6** (DINOV2) | 64.3 (Gemini) | ±0.3 |
| Audio | CLAP | ESC | 2 | **91.3** | **90.5** (CLAP) | - | ±1.2 |
|       |      | RAV | 1.4 | 38.4 | **47.9** (Whisper) | - | ±2.5 |
|       |      | US8 | 8.7 | **79.8** | 77.1 (CLAP) | - | ±0.8 |
| Biological | BioCLIP2 | FSH | 94 | 80.2 | **83.7** (BioCLIP) | 59.5 (Gemini) | ±0.3 |
|            |          | AWA | 37 | 95.7 | **97.1** (BioCLIP) | 96.5 (Gemini) | ±0.2 |
|            |          | PLD | 2.5 | 67.4 | **72.0** (BioCLIP) | **74.2** (Gemini) | ±1.8 |
| Tabular | TabPFNv2 | CC18 | 155 | 84.5 | **87.8** (TabPFNv2) | 50.1 (TabLLM-Gemini) | ±0.2 |
|         |          | R25 | 35 | 66.0 | **67.0** (TabPFNv2) | 05.1 (JOLT-Qwen-2.5-3B) | ±0.5 |
| **(Score, # Models)** | | | - | (78.9, 1) | (81.4, 5) | (62.2, 4) | - |

count via a grid search. The LLM / VLM baseline results in the paper are reported using the best performing LLM / VLM in the class (we consider QwenVL 2.5 3B Instruct and Gemini-Flash-2.0 via the Gemini API). All MARVIS results are zero-shot in the sense that we do not give examples of the task to the VLM at inference time; they are full-shot in the sense that the embedding-generating models have access to the entire test set without labels. For the LLM / VLM baselines, image classification is performed zero-shot. Tabular classification and regression uses the JOLT (Shysheya et al., 2025) and TabLLM (Hegselmann et al., 2023) strategies with k-shot computed dynamically based on the maximum context length. We report the best result in the table. Specialist models are full-shot, and we report the best overall result in the table. For extended results, a detailed description of the method we use to generate our novel tabular benchmarks CC18-Semantic and Regression2025-Semantic, and a deeper dive into tabular data, including balanced metrics, please refer to Appendix G.

**Specialized model baselines.** For vision, the best performing specialist was the large DinoV2 model with a registry and KNN classification (Oquab et al., 2023). For audio, the CLAP model with contrastive zero-shot classification from Microsoft and OpenAI's Whisper-V2-Large model with KNN classification perform the best (Radford et al., 2022; Elizalde et al., 2023; Ma et al., 2024a). For biological data, BioCLIPv2 with KNN classification performs the best (Gu et al., 2025). For tabular data, TabPFNv2 with standard forward pass classification and regression is a strong baseline; we also consider classical baselines such as CatBoost and linear models in Appendix G (Prokhorenkova et al., 2018; Hollmann et al., 2025).

**LLM / VLM baselines.** For vision, we use the standard strategy of zero-shot prompting and exact match extraction described in works such as (Zhang et al., 2024). For audio, we are unable to compare to public API-based models, as to the best of our knowledge, no generalist exists capable of performing audio classification.

**LLM tabular baselines.** In the tabular domain, as a secondary contribution, we generate the first large-scale standardized benchmarks for tabular classification and regression that include semantic class names, feature names and metadata; CC18-Semantic and Regression 2025 Semantic. We also re-implement two prominent

LLM-tabular methods, TabLLM and JOLT (Hegselmann et al., 2023; Shysheya et al., 2025), which lack general-purpose implementations. For more details on this, please refer to Appendix G.

**Additional details.** For more analysis on the embedding models and baselines, please refer to Appendix B. For more explanation of the benchmarks we use, please refer to Appendix A.

### 3.1 FINDINGS

**MARVIS is competitive with SOTA specialist predictors.** Across a wide range of modalities, we observe that MARVIS strongly conserves predictive performance – across most tasks we consider, it is able to match the best specialist model in the cohort. By comparison, the best existing LLM / VLM methods, tailored for each domain, achieve 77% of specialist performance on average. Remarkably, we find that MARVIS is a more accurate image classifier than Gemini Flash 2.0, despite never actually having seen the images. MARVIS also sometimes improves on specialists; it outperforms CLAP, a specialist contrastive predictor, using its own embeddings.

> **Contributions**
>
> MARVIS-3B achieves competitive performance across four distinct modalities, approaching and occasionally exceeding the best specialist predictors, and improving on LLM / VLM-only methods by 16.7%.

**MARVIS outperforms direct fine-tuning of its base model.** In section D, we describe a novel method for fine-tuning an LLM directly on the embeddings of an upstream model such as TabPFNv2. We test this method (Qwen-FFT) at inference time and find that it is highly accurate, far outperforming previously published strategies such as JOLT and TabLLM for general-case tabular inference with LLMs; however, in section E.2, we show that MARVIS-3B outperforms even this strong baseline on average.

**VLMs reason over their input data and condition their behavior based on the context provided.** One core research question, from our perspective, was whether a VLM was simply copying learned patterns or utilizing simple heuristics to achieve this strong performance. Systematic analysis of VLM reasoning in Fig. 3 demonstrates clear correlations between reasoning quality and metric gains, on average, across three tabular classification datasets (two with meaningful semantic features, one without).

Further analysis of disagreement patterns reveals that only 35% of methods agree on all test cases, with 65% showing partial disagreement. Furthermore, in Table 2, we show that different visualization methods elicit systematically different reasoning approaches, providing strong evidence that VLMs adapt their analysis based on visual information content. Still more evidence can be found in Appendix I.1. We observe that different visualization methods elicit systematically different reasoning approaches, providing strong evidence that VLMs adapt their analysis based on the available visual information. **tsne_knn** produces quantitative neighbor analysis with explicit distance calculations (average 48.0 words), **tsne_semantic_axes** integrates semantic class information with spatial reasoning (304.9 character responses) and **tsne_perturbation_axes** generates the longest, most detailed responses (310.6 characters) with sophisticated uncertainty analysis. These patterns suggest that VLMs engage in more thorough spatial analysis when the visual information supports accurate classification, indicating genuine reasoning rather than pattern matching.

The systematic variation in reasoning style directly correlates with the information content of each visualization method, demonstrating that VLMs genuinely process and respond to different types of visual information. Detailed analysis of these reasoning patterns and their implications for VLM spatial understanding is provided in Appendix I.

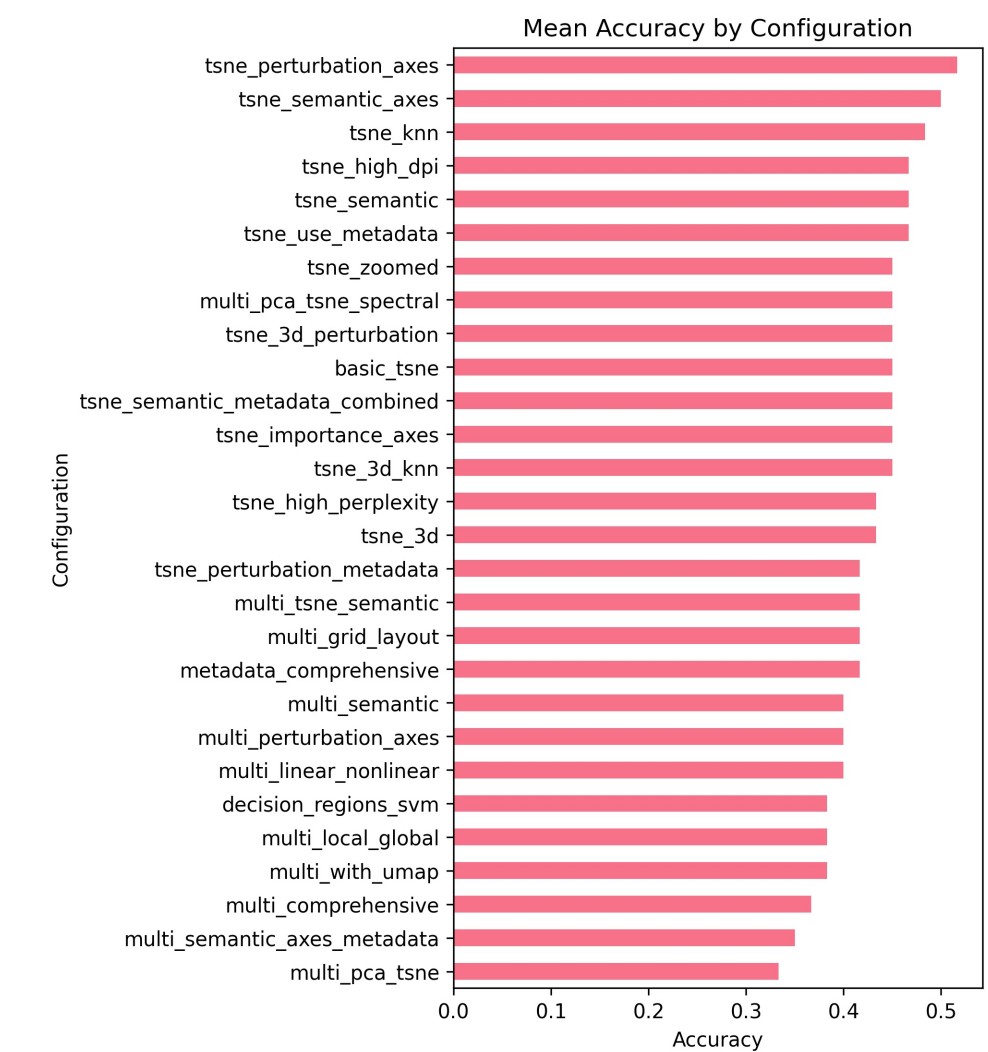

Figure 3: **The selection of context strongly influences MARVIS performance.** We ablate over twenty different context composition strategies, and find that perturbation-based approaches with uncertainty analysis achieve the highest performance, followed by semantic axes with meaningful class labels. The majority of the experiments in the paper are conducted using TSNe + KNN, because it exposes less information about the underlying data and therefore better reflects real-world use.

**The flexibility of MARVIS allows for more complex use cases.** In Fig. 4, we demonstrate one such use case – open-ended chat about a particular predictive result. In this example, the user asks MARVIS to assess its own performance and recommend strategies to improve results in the future.

Table 2: **Method-Specific Reasoning Patterns.** Each visualization method elicits distinct reasoning behaviors: k-NN methods trigger quantitative distance analysis, perturbation methods generate longer responses, and basic methods rely heavily on proximity heuristics. Here, Resp. Length refers to the token count of responses, distance mentions to the rate at which the response mentions distance between points in embedded space, and closest usage refers to how often MARVIS uses the word "closest" in its response.

| Method | Resp. Length | Distance Mentions | Closest Usage |
|---|---|---|---|
| tsne_3d_perturbation | **365.3** | 0.000 | 0.433 |
| tsne_perturbation_axes | 310.6 | 0.000 | **0.650** |
| tsne_semantic_axes | 304.9 | 0.000 | 0.683 |
| tsne_knn | 279.0 | **0.650** | 0.883 |
| basic_tsne | 268.3 | 0.000 | 1.000 |

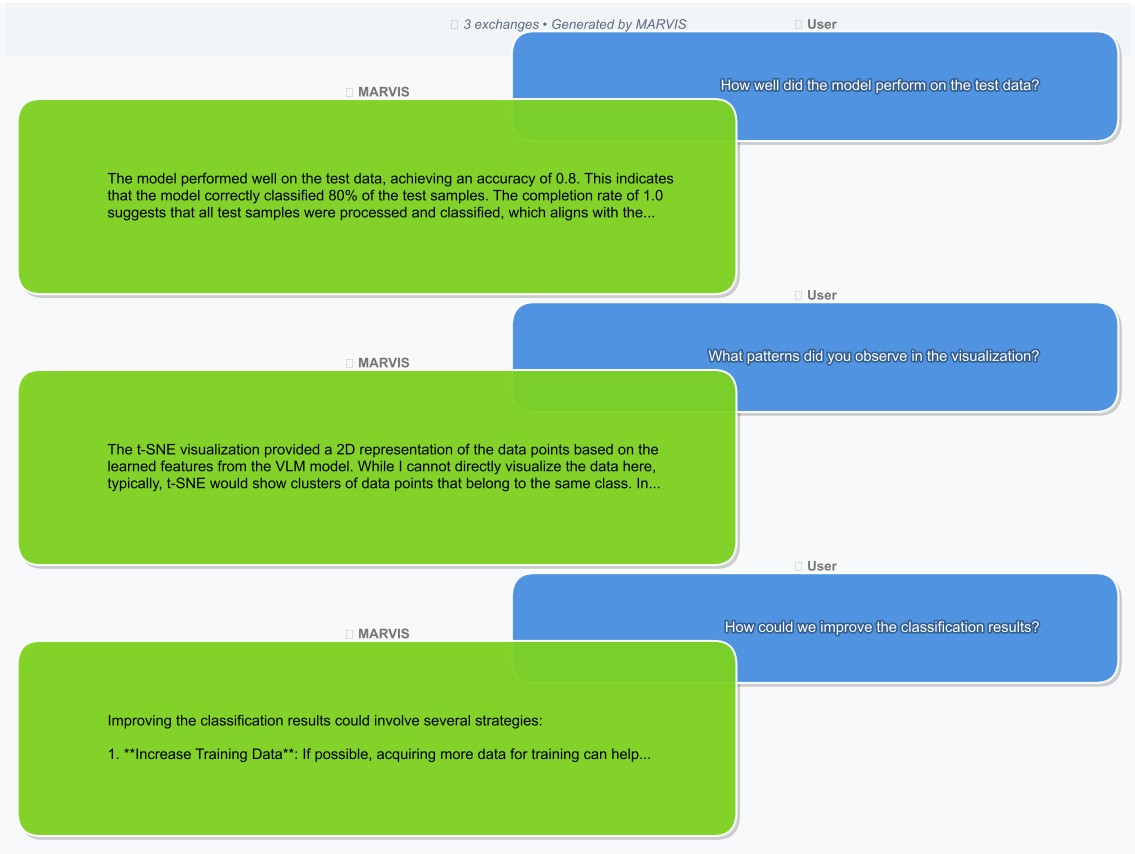

Figure 4: **MARVIS extends traditional predictive capabilities.** Because it requires no fine-tuning, and because it exposes the VLM's classification process to the VLM itself, MARVIS enables VLMs to reason over, and converse about, their predictive performance.

## 4    RELATED WORK

MARVIS builds on extensive prior work in vision-language models (VLMs) which has followed two primary evolutionary tracks: maximalist approaches from industry labs focusing on peak performance, and minimalist open-source approaches prioritizing efficiency and accessibility; in Appendix F, we trace the history of this evolution in greater detail.

The use of embedding spaces for cross-modal understanding has roots in representation learning (Bengio et al., 2013) and dimensionality reduction techniques (Van der Maaten & Hinton, 2008). Recent work has explored the geometric properties of embedding spaces (Ethayarajh, 2019) and their visualization for interpretability (Liu et al., 2017). t-SNE and UMAP have been widely used for visualizing high-dimensional data (McInnes et al., 2018), but their application to VLM reasoning represents a novel paradigm. Previous work on visual reasoning has focused on spatial relationships in natural images (Johnson et al., 2017), but MARVIS extends this to abstract embedding spaces across arbitrary modalities.

MARVIS distinguishes itself from existing approaches through several key innovations: (1) **Training-free adaptation**: Unlike approaches requiring extensive fine-tuning, MARVIS leverages pre-trained components without modification; (2) **Universal modality support**: A single architecture handles any data type through embedding visualization; (3) **Privacy preservation**: Visualization of embeddings avoids raw data exposure; (4) **Computational efficiency**: Achieves competitive performance with a 3B parameter model versus much larger specialized systems.

## 5    CONCLUSION

We introduce MARVIS, a training-free method that enables small VLMs to predict across any data modality through embedding visualization. By transforming embedding spaces into visual representations optimized for VLM spatial reasoning, MARVIS achieves competitive performance across diverse domains.

MARVIS addresses key limitations in existing approaches: it requires no domain-specific training, preserves data privacy through visualization rather than serialization, and maintains competitive performance. The approach demonstrates that visual reasoning can serve as a universal interface for foundation models across any data modality.

Based on this, we propose several key principles for designing effective VLM interfaces:

- **Information density matters**: Richer visualizations elicit more sophisticated reasoning
- **Method-purpose alignment**: Different visualization approaches suit different reasoning tasks
- **Adaptive interface design**: VLMs can effectively utilize different types of visual information

Future work includes further investigation of the optimal mix of visualizations and embeddings to boost performance and fine-tuning strategies which may improve the performance of base VLMs for reasoning over scientific imagery, including reasoning post-training.

## REPRODUCIBILITY STATEMENT

We have, to the best of our ability, ensured that all experiments described in this paper are reproducible in principle. In order to facilitate this, we provide an anonymized source code repository containing the exact training/evaluation orchestration used in our experiments, including the OpenML CC18 runner, evaluation harness, baseline integrations, and analysis scripts. All datasets, splits, and preprocessing steps for CC18 are clearly documented (including feature selection choices and filters). Exact hyperparameters, seeds, and

evaluation metrics are summarized in our Appendix. Finally, we will release archives of raw predictions and per-dataset metrics for post hoc verification.

ETHICS STATEMENT

MARVIS enhances privacy preservation in machine learning by avoiding raw data serialization, instead using anonymized embedding visualizations. This approach reduces risks of data exposure while maintaining model performance. The method's universal applicability could democratize access to advanced ML capabilities across diverse scientific domains.

LLM USE STATEMENT

In accordance with ICLR policy, the authors acknowledge the limited use of LLMs for generating code and LaTeX, rendering visualizations, and polishing writing.

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

CONTENTS

## A    APPENDIX: BENCHMARK DATASET DESCRIPTIONS

### A.1    VISION BENCHMARKS

**CIFAR-10**: One of the most widely used datasets for computer vision research: contains 60,000 32×32 color images in 10 classes (airplanes, cars, birds, cats, deer, dogs, frogs, horses, ships, trucks) with 6,000 images per class. Split into 50,000 training and 10,000 test images Krizhevsky (2009).

**CIFAR-100**: Similar to CIFAR-10 but with 100 classes containing 600 images each (500 training, 100 test per class). The 100 classes are grouped into 20 superclasses, making this a more challenging classification benchmark.

### A.2    AUDIO BENCHMARKS

**ESC-50 (Environmental Sound Classification)**: Contains 2,000 environmental audio recordings with 50 classes and 40 clips per class. Each clip is 5 seconds long at 44.1 kHz, single channel, extracted from public field recordings through Freesound.org Piczak (2015).

**RAVDESS (Ryerson Audio-Visual Database of Emotional Speech and Song)**: Audio dataset focusing on emotion recognition tasks, commonly used for evaluating emotional speech and song recognition capabilities Livingstone & Russo (2018).

**UrbanSound8K**: Contains 8,732 labeled sound excerpts with 10 classes of outdoor/urban sounds, specifically designed for benchmarking sound classification models in urban environments.

### A.3    BIOLOGICAL/SCIENTIFIC VISION BENCHMARKS

**FishNet**: Large-scale dataset with 94,532 images from 17,357 aquatic species, organized by biological taxonomy (8 classes, 83 orders, 463 families, 3,826 genera). Includes bounding box annotations and supports classification, detection, and functional trait prediction tasks Khan et al. (2023). We treat FishNet as a classification problem over families.

**AWA2 (Animals with Attributes 2)**: Animal classification dataset used for zero-shot learning tasks, focusing on learning representations with animal attributes. Part of challenging benchmarks alongside CUB and SUN datasets Xian et al. (2019). We treat AWA2 as a 50-class classification problem with no holdout classes.

**PlantDoc**: Contains 2,569 images across 13 plant species and 30 classes (diseased and healthy) with 8,851 total labels. Split into 2,328 training and 237 test images, with unbalanced classes ranging from 50-180 images per class Singh et al. (2020).

### A.4    TABULAR BENCHMARKS

**OpenML CC18**: Curated benchmark suite of 72 classification datasets from OpenML 69 of which we utilize), selected based on strict criteria:

- Size: 500-100,000 observations, $\leq$ 5,000 features
- Quality: No artificial data, minority/majority class ratio $\geq$0.05
- Usability: Compatible with multiple algorithms, representing commonly used ML datasets

See Bischl et al. (2021) for more on this benchmark, including the complete specification of tasks.

**Regression 2025**: Custom benchmark of 43 regression tasks from 2015-2025 sourced from OpenML, evaluated using $R^2$ scores on a 0-100 scale for consistent comparison across tasks; introduced onto the OpenML plat-

form in March 2025 at openml.org/search?type=benchmark&sort=tasks_included&study_type=task&id=455. Please follow the link for the complete list and specification of tasks. After discarding tasks on which all models fail, we compute our scores on a subset of 33.

## B  IMPLEMENTATION DETAILS

This section contains additional experimental details from the paper.

### B.1  EMBEDDING MODELS

**Vision**: DINO-v2-ViT-L-14-reg provides robust visual representations trained through self-supervised learning on large-scale image datasets Oquab et al. (2023).

**Audio**: Microsoft CLAP employs contrastive audio-language pre-training to create joint embeddings for audio and text modalities Elizalde et al. (2023).

**Biological**: BioCLIP2 specializes in scientific vision understanding, trained on biological image-text pairs for enhanced performance on scientific datasets. It is the latest in a series of foundation models for biological applications, initiated by BioCLIP, which incorporated taxonomic labels in the vision-language contrastive training, yielding promising species classification accuracy Stevens et al. (2024). Follow-up work scaled data to 162M images (BioTrove, Yang et al., 2024), specialized the data to camera traps (CATALOG and WildCLIP, Gabeff et al., 2024; Santamaria et al., 2025), and added additional model modalities (TaxaBind, Sastry et al., 2025).

**Tabular**: Tabular machine learning has traditionally relied on specialized approaches including tree-based methods (Random Forest Breiman (2001), XGBoost Chen & Guestrin (2016), CatBoost Prokhorenkova et al. (2018)) and specialized neural architectures (TabNet Arik & Pfister (2021), TabTransformer Huang et al. (2020)). TabPFN Hollmann et al. (2022) employed transformer-based in-context learning, and was later extended to support larger datasets Feuer et al. (2024); Hollmann et al. (2025); Müller et al. (2025). In this work, we use TabPFNv2 as our embedding generating model.

### B.2  HYPERPARAMETERS

In this section, we document the hyperparameters used for our main experiments section.

**t-SNE Configuration**:

- Perplexity: 15 (optimized through ablation studies)
- Iterations: 1000 for stable convergence
- Learning rate: 200 (default)
- Random state: Fixed for reproducibility

**KNN Configuration**

- nn = 30
- metric = 'euclidean' (general), 'cosine' (embeddings)
- weights = 'distance'

**Tabular Baseline Models Configuration**:

**CatBoost (Classification & Regression)**

- iterations: 1000
- depth: 6
- learning_rate: 0.03
- random_seed: 42
- verbose: False
- Categorical features: Auto-detected and preserved

**TabPFN v2 (Classification & Regression)**

- n_estimators: 8
- device: Auto-detected (CUDA if available)
- ignore_pretraining_limits: True
- Target preprocessing: Quantile binning for regression
- Max quantiles: min(n_samples // 2, 1000)
- NaN/INF imputation: Median strategy

**Random Forest (Classification & Regression)**

- n_estimators: 100
- max_depth: None (unlimited)
- random_state: 42
- n_jobs: -1 (all cores)

**Gradient Boosting (Classification & Regression)**

- n_estimators: 100
- learning_rate: 0.1
- random_state: 42
- Feature selection: Max 500 features (SelectKBest)

**Logistic/Linear Regression**

- max_iter: 1000 (Logistic only)
- C: 1.0 (Logistic regularization)
- random_state: 42
- n_jobs: -1 (all cores)
- Preprocessing: StandardScaler applied

## C  COMPUTATIONAL EFFICIENCY

**Model Size**: MARVIS uses Qwen2.5-VL (3B parameters).

**Inference Time**: Average processing time per sample ranges from 0.5-2.0 seconds depending on visualization complexity and VLM reasoning depth.

**Memory Requirements**: All experiments are conducted using 1xH100 80GB GPUs on a hosted Lambda cluster. Peak memory usage remains under 8GB GPU memory for batch processing, enabling deployment on standard hardware.

**GPU Utilization**: For development and testing combined, we estimate 1,500 H100-hours were used during the creation of this paper.

## D  FULL FINETUNING EXPERIMENTS

As a strong baseline for MARVIS, we introduce a novel approach to LLM fine-tuning, projecting a sequence of positionally encoded TabPFNv2 embeddings and learned label tokens into the model's token space. At inference time, we project the test element embedding from TabPFNv2 into the model's token space and conduct standard autoregressive inference to acquire the predicted label.

### D.1  BALANCED PREFIX CONSTRUCTION

We construct a balanced, few-shot prefix from training embeddings using `prepare_tabpfn_embeddings_for_prefix`. Given class labels $y$ and train embeddings $E \in \mathbb{R}^{N \times d}$ (after robust scaling and optional resizing), we select a total of `num_few_shot_examples` examples across classes, distributing as evenly as possible; short classes are repeated to meet demand. The resulting prefix tensor $P \in \mathbb{R}^{M \times d}$ (with class labels $c \in \{0, \dots, K-1\}^M$) is saved to `prefix_data.npz`.

### D.2  SPECIAL TOKENS AND CLASS TOKENS

We extend the tokenizer with two sentinel tokens `<PREFIX_START>` and `<PREFIX_END>` and with up to 10 class tokens `<CLASS_i>`. The underlying embedding matrix is resized accordingly. These token IDs delimit the region where external embeddings will be injected and provide stable referents for class-conditional evidence tokens.

### D.3  POSITION-WISE PROJECTION INTO TOKEN SPACE

**Implementation.**  The core mechanism is implemented via `QwenWithPrefixEmbedding`:

- A learnable projector is defined as `Linear(d, H)`, mapping TabPFNv2 embedding dimension $d$ to the LLM hidden size $H$.
- During `forward`, we build `inputs_embeds` from `input_ids` and locate the span between `<PREFIX_START>` and `<PREFIX_END>`. Let the number of available positions be $T$.
- If embeddings and class labels are provided, we compute $\tilde{P} = PW + b \in \mathbb{R}^{M \times H}$ and interleave with class token embeddings: even positions receive projected vectors, odd positions the embeddings of `<CLASS_{c_j}>`, truncated to $T$.
- If only embeddings are provided, we fill the $T$ positions with $\tilde{P}$ contiguously.
- The modified `inputs_embeds` are passed to the base model with `input_ids=None`.

**Rationale and soundness.**

1. **Representation Alignment.** A learned affine map is the minimal adapter aligning TabPFN geometry to the LLM token manifold, akin to prefix/prompt-tuning adapters.
2. **Token-Sequential Semantics.** Injecting a bounded token span leverages positional mixing and attention for fusion with the downstream textual prompt; class-token interleaving ties directions in $\tilde{P}$ to discrete label anchors.

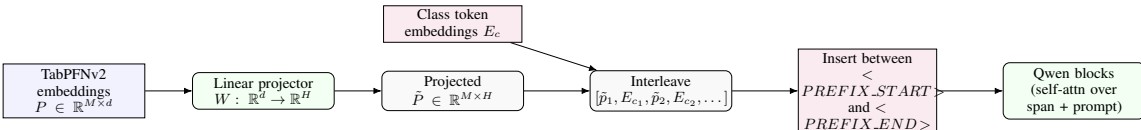

Figure 5: **Projection and interleaving of TabPFNv2 embeddings into the LLM token space.**

3. **Identifiability.** With only the projector and last $k$ layers unfrozen, gradients supervise a compact subspace, preserving language priors while enabling consistent task adaptation. Another parameter-efficient approach which we do not consider in this draft, LORA, would likely produce similar outcomes.

## D.4 BACKBONE AND HOOKS

The default backbone is `Qwen/Qwen2.5-3B-Instruct` (configurable via `--model_id`). MARVIS prepares the model with prefix-embedding tokens and class tokens using `prepare_qwen_with_prefix_embedding`. Optional Vector Quantization (VQ) is available via `prepare_qwen_with_vq_prefix_embedding`.

## D.5 LABEL ENCODING

We encode labels with a `LabelEncoder` fitted on train+val+test labels per task; IDs index into the class token set. For float labels near-integral, we cast to integers; otherwise, regression handling is separate.

## D.6 FFT TRAINING CONFIGURATION

We train using `train_llm_with_tabpfn_embeddings`. Key elements:

- **Backbone freezing:** Unfreeze the last $k$ layers (default $k{=}1$) and the projector; other layers frozen.
- **Loss:** Cross-entropy over class-token targets in the output; attention integrates projected evidence with the prompt.
- **Optimization:** Defaults: `batch_size=8`, `grad_accum_steps=1`, `total_steps=2000`, `save_steps=500`, `lr=1e-4`, `mixup_alpha=0.0`, early stopping (patience 30, threshold 0.4).
- **Prefix length:** Template ensures enough positions between `<PREFIX_*>`; excess prefix entries are truncated.
- **W&B:** Enabled with dated project names for versioning; run names encode task/split.

## D.7 FFT EVALUATION PROTOCOL

Evaluation is handled by `examples/tabular/evaluate_on_dataset_tabular.py` with the unified `--models` interface. The orchestrator passes the saved model directory and, unless `--no_baselines` is set, appends `all_baselines`.

- Test size limit: We commonly use `--max_test_samples 200` to cap test evaluation for rapid iteration.
- Feature selection threshold: `--feature_selection_threshold` can be forwarded for high-dimensional datasets.
- Metrics and artifacts: Saved under each task/split `evaluation` directory and logged to W&B.

### D.8 FFT LIMITATIONS AND DISCUSSION

While, for the sake of having strong reasonable baselines, we include this approach, we believe that in practice, it is not a suitable general-purpose substitute for MARVIS.

- **Fine-tuning degrades chat performance.** By changing the VLM's vocabulary and last $k$ layers, we necessarily degrade chat performance somewhat; this weakens one of the major use cases for MARVIS.

- **Fine-tuning degrades interpretability.** Because the VLM does not "know" it was fine-tuned on the data, nor does it "know" what it learned during fine-tuning, it cannot reason nearly as effectively about its own decision-making process, weakening another major use case for MARVIS.

- **Fine-tuning must be done again for every new dataset.** This is an inconvenience as it requires the end user to maintain suitable training infrastructure on top of their pure inference infrastructure, which is generally more flexible.

## E  EXTENDED RESULTS

### E.1  ABLATION STUDY ON CONTEXT CHOICE DETAILS

For a list of the methods we consider, please refer to Table 3.

Extended ablation studies reveal optimal configurations across different visualization strategies. We systematically evaluated four key approaches to understand how different types of information affect VLM spatial reasoning performance.

The configuration performance hierarchy demonstrates clear patterns:

- **tsne_perturbation_axes**: 51.7% accuracy with uncertainty analysis

- **tsne_semantic_axes**: 50.0% accuracy with meaningful class labels

- **tsne_knn**: 48.3% accuracy with explicit neighbor information

- **basic_tsne**: 45.0% accuracy as baseline approach

#### E.1.1  ANALYSIS OF CONFIGURATION EFFECTS

The ablation results reveal several key insights about VLM spatial reasoning:

**Perturbation-based Enhancement**: The tsne_perturbation_axes configuration achieves the highest performance by incorporating uncertainty information through small perturbations around the query point. This provides the VLM with richer spatial context about decision boundaries and confidence regions.

**Semantic Information Value**: The tsne_semantic_axes approach shows strong performance by providing meaningful class labels within the visualization. This allows the VLM to leverage both spatial relationships and semantic understanding simultaneously.

**Neighbor Information Benefits**: The tsne_knn configuration demonstrates moderate improvements over the baseline by explicitly highlighting nearest neighbors, helping the VLM focus on locally relevant information.

**Baseline Robustness**: Even the basic_tsne approach achieves reasonable performance (45%), validating the fundamental effectiveness of the visual reasoning paradigm across modalities.

| Category | Method | Description |
|---|---|---|
| Basic Visualizations | basic_tsne | Standard t-SNE visualization with default parameters |
| | tsne_3d | Three-dimensional t-SNE visualization for enhanced spatial understanding |
| | tsne_high_dpi | High-resolution t-SNE with increased image quality |
| | tsne_high_perplexity | t-SNE with modified perplexity parameter for different clustering |
| Enhanced Single Methods | tsne_knn | t-SNE with k-nearest neighbor information overlay |
| | tsne_perturbation_axes | t-SNE with perturbation analysis for uncertainty quantification |
| | tsne_semantic_axes | t-SNE with semantic class labels and axes descriptions |
| | tsne_3d_knn | 3D t-SNE visualization with k-NN connections displayed |
| | tsne_3d_perturbation | 3D t-SNE with perturbation analysis for spatial uncertainty |
| Multi-Visualization Methods | multi_comprehensive | PCA + t-SNE + Spectral + Isomap comprehensive view |
| | multi_pca_tsne | Combined PCA and t-SNE dual visualization |
| | multi_pca_tsne_spectral | Triple visualization: PCA + t-SNE + Spectral embedding |
| | multi_linear_nonlinear | Linear and nonlinear dimensionality reduction comparison |
| | multi_local_global | Local and global structure preservation methods |
| | multi_with_umap | Multi-method visualization including UMAP |
| | multi_grid_layout | Grid-based layout for systematic method comparison |
| Specialized Methods | decision_regions_svm | SVM decision boundary visualization with regions |
| | frequent_patterns | Pattern mining visualization for feature relationships |
| | metadata_comprehensive | Metadata-enhanced comprehensive visualization approach |

Table 3: **MARVIS Method Variants Overview.** Comprehensive summary of visualization approaches evaluated in ablation studies, categorized by methodology type and complexity level.

### E.2 ABLATION ON MARVIS BACKEND AND FFT

This ablation (tabular classification on a subset of the entire OpenML CC-18 Semantic benchmark) indicates that MARVIS's base performance depends considerably more on the choice of embedding generating model than on the choice of VLM backend; a small QwenVL 2.5 3B model (MARVIS_3B) outperforms a more recent thinking model (moonshotai/Kimi-VL-A3B-Thinking-2506 referenced as MARVIS_kimi) and matches GPT-4o-mini (MARVIS_gpt4o). MARVIS-3B also outperforms the full fine-tuning solution described in section D by a substantial margin (Qwen-FFT in the figure); although the FFT solution generally is able to reduce loss to near-zero on the training data, it sometimes fails to generalize well, particularly when the training dataset size is small.

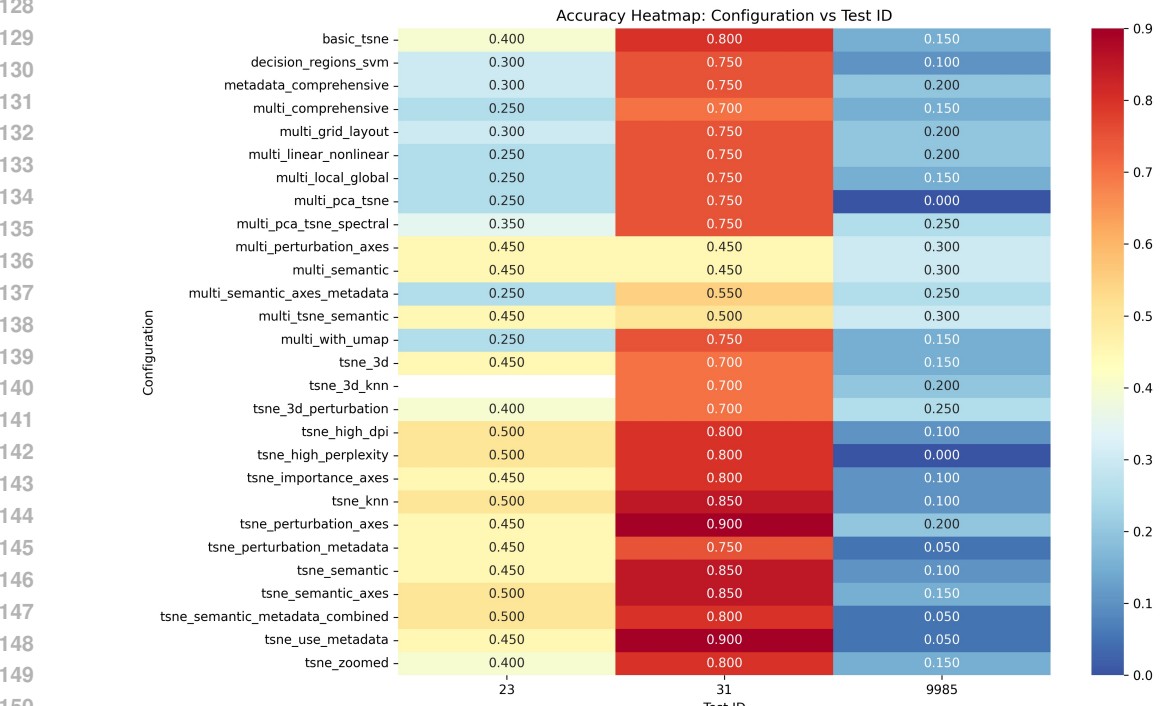

Figure 6: **Configuration Performance Heatmap.** Detailed breakdown showing performance variations across different parameter combinations and visualization strategies. Darker regions indicate higher accuracy, with perturbation-based methods consistently showing superior performance across various settings.

## F    EXTENDED RELATED WORKS

Early VLM architectures explored complex fusion mechanisms to achieve deep integration between vision and language. Flamingo (Alayrac et al., 2022) introduced gated cross-attention layers interleaved within frozen LLMs, enabling few-shot learning across diverse multimodal tasks without task-specific fine-tuning. BLIP (Li et al., 2022) and its successor BLIP-2 (Li et al., 2023b) pioneered the Multimodal Mixture of Encoder-Decoder (MED) architecture and introduced the Q-Former as a lightweight bridge between frozen vision encoders and language models. PaLI (Chen et al., 2022) established the principle of joint scaling, demonstrating that optimal VLM performance requires balanced scaling of all components: vision models, language models, and training data.

LLaVA (Liu et al., 2023a) democratized VLM research by establishing an efficient, open-source blueprint. Its three-component architecture—frozen vision encoder, lightweight MLP projector, and frozen LLM—with two-stage training (feature alignment followed by instruction tuning) proved that simple architectures could achieve impressive multimodal capabilities. LLaVA-NeXT (Liu et al., 2024) introduced dynamic high resolution through intelligent image partitioning, while mPLUG-Owl2 (Ye et al., 2023) developed Modality-Adaptive Modules to foster positive cross-modal collaboration while mitigating interference. POINTS (Ma et al., 2024b) exemplified sophisticated data curation through perplexity-based filtering.

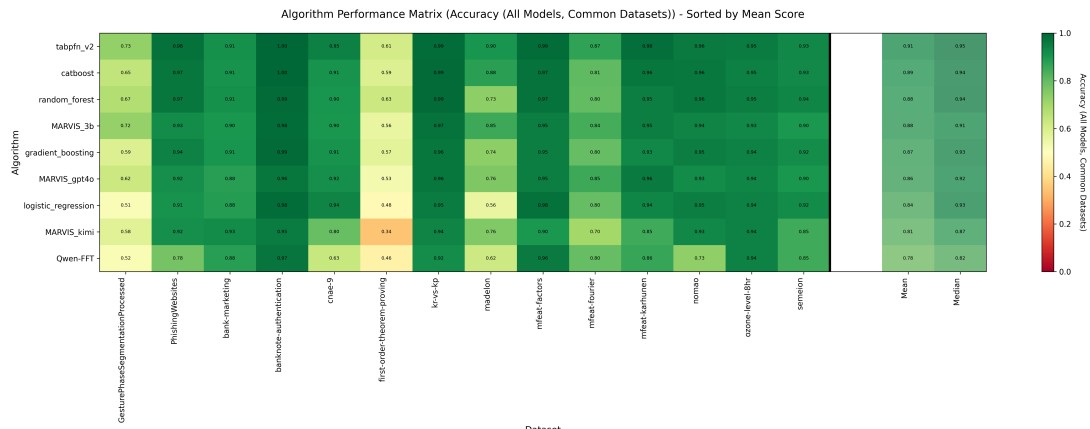

Figure 7: **Accuracy matrix for MARVIS backend variants and FFT.** Our ablation shows that MARVIS's base performance depends considerably more on the choice of embedding generating model than on the choice of VLM backend; a small QwenVL 2.5 3B model outperforms a more recent thinking model and matches GPT-4o-mini.

Recent work has pushed beyond conversational capabilities toward precise, spatially-grounded understanding, key to understanding the gains in MARVIS. Grounding DINO (Liu et al., 2023b) achieved open-set object detection through text-conditioned spatial understanding, while KOSMOS-2 (Peng et al., 2023) integrated coordinate tokens directly into the LLM vocabulary for grounded text generation. OtterHD (Li et al., 2023a) pioneered an encoder-less architecture, processing raw pixel patches directly in the LLM to eliminate resolution constraints. SleighVL (Liu et al., 2025) refined high-resolution processing through attention-based sub-image weighting via Global Semantic-guided Weight Allocation. Emu3 (Wang et al., 2024) unifies vision and language modalities under next-token prediction, tokenizing images, videos, and text into a shared vocabulary space. Molmo (Deitke et al., 2024) champions fully open ecosystems with human-annotated data, breaking dependence on proprietary synthetic datasets. Early cross-modal strategies used feature concatenation, attention mechanisms, or late fusion strategies, requiring extensive retraining for each new modality (Baltrusaitis et al., 2018). Modern paradigms include contrastive learning (CLIP-style) (Radford et al., 2021), generative modeling (Ramesh et al., 2022), and instruction tuning (Wei et al., 2022). However, these approaches typically require substantial computational resources and domain-specific training data for each new modality.

# G    DEEP DIVE: TABULAR MODALITY ANALYSIS

This section provides a comprehensive analysis of MARVIS performance on tabular data, evaluating both classification and regression tasks against established baselines. The analysis includes detailed performance metrics, correlation studies with TabPFN v2, and critical difference plots for statistical comparison.

## G.1    BASELINES: JOLT AND TABLLM

One challenge we faced during the creation of this paper is that prior work which utilized LLMs for tabular classification and regression lacked both standard benchmarks and consistent, easy to implement methods. As a secondary contribution, we release comprehensive full-size tabular benchmarks which include semantic information (see H), and modern, feature-complete implementations of TabLLM and JOLT.

**Dual Implementation Architecture:** We developed a sophisticated dual-path architecture that supports both legacy compatibility and modern framework integration. Our implementation includes:

- **Legacy Integration:** Direct incorporation of original JOLT codebase with automatic fallback mechanisms
- **Modern Implementation:** Complete HuggingFace transformers integration with VLLM backend support
- **Unified Model Loader:** Centralized model management supporting multiple backends (HuggingFace, VLLM, OpenAI, Gemini)

**Memory Optimization and Scalability:** Critical for production deployment, our implementation includes:

- Gradient checkpointing with KV cache disabling for memory efficiency
- Dynamic batch sizing with automatic Out-of-Memory (OOM) recovery
- Aggressive memory limits for regression tasks (512MB default)
- Feature dropping with retry mechanisms for large datasets

**Enhanced Task Support:** Beyond the original classification focus, we extended JOLT to support:

- Full regression pipeline with intelligent binning strategies
- Automatic task type detection and configuration
- Balanced few-shot example selection algorithms
- Context-aware prompt truncation for varying model context lengths

**Configuration Management:** We developed a comprehensive metadata system:

- Automatic JOLT configuration discovery by OpenML task ID
- Feature count validation ensuring dataset-configuration alignment
- Semantic feature mapping from original to descriptive names
- Graceful degradation when configurations are unavailable

**TabLLM Implementation**

**Real-time Note Generation:** Our TabLLM implementation eliminates the need for pre-generated note banks through:

- On-the-fly natural language description generation
- Dynamic semantic feature expansion matching actual dataset characteristics
- Template-based prompt generation with YAML configuration support
- Automatic feature alignment verification post-preprocessing

**Multi-Backend API Support:** We created a unified interface supporting:

- OpenAI API integration (GPT-4, GPT-3.5-turbo, GPT-4o)
- Google Gemini API support with automatic model selection

- Local model deployment via HuggingFace transformers
- Automatic backend detection based on model naming conventions

**Quality Assurance Mechanisms:** To ensure generation quality, we implemented:

- Inspection system saving sample generated notes for manual review
- N-gram analysis for content validation and diversity assessment
- Context truncation with intelligent few-shot example selection
- Template validation ensuring prompt completeness

### HuggingFace Ecosystem Compatibility

Both implementations leverage the complete HuggingFace ecosystem:

- `AutoModelForCausalLM` and `AutoTokenizer` for model loading
- Trust remote code support for cutting-edge models
- Automatic device placement and memory optimization
- Support for quantized models (8-bit, 4-bit) through BitsAndBytes

### VLLM Integration

For production deployments requiring high throughput:

- Automatic VLLM backend selection for compatible models
- Tensor parallelism configuration for multi-GPU deployment
- Optimized sampling parameters with fallback to transformers
- Unified generation interface across backends

### Benchmark Integration

Our implementations integrate seamlessly with standard evaluation frameworks:

- Direct OpenML dataset loading and preprocessing
- Standardized evaluation interface compatible with scikit-learn
- Comprehensive metrics calculation (accuracy, F1, ROC-AUC, R², MAE, MSE)
- Weights & Biases integration for experiment tracking

### Usage and Accessibility

Our implementations provide simple, unified interfaces:

```
# JOLT evaluation with local model
python examples/tabular/evaluate_llm_baselines_tabular.py \
    --models jolt \
    --dataset_ids 23 \
    --jolt_model Qwen/Qwen2.5-7B-Instruct

# TabLLM evaluation with API backend
python examples/tabular/evaluate_llm_baselines_tabular.py \
```

```
    --models tabllm \
    --dataset_ids 1590 \
    --openai_model gpt-4o
```

This unified interface abstracts away implementation complexity while providing extensive configuration options for advanced users.

## G.2    CLASSIFICATION PERFORMANCE ON OPENML CC18

The OpenML CC18 benchmark represents one of the most comprehensive evaluation suites for tabular classification, consisting of 72 carefully curated datasets Bischl et al. (2021).

| Model | Mean Acc. | Balanced Acc. | F1 Macro | Datasets |
|---|---|---|---|---|
| MARVIS | 84.5% | 80.2% | 79.9% | 69 |
| TabPFN v2 | **87.8%** | **82.2%** | **82.3%** | 66 |
| CatBoost | 87.0% | 81.5% | 81.8% | 70 |
| Random Forest | 86.5% | 80.3% | 81.0% | 70 |
| Gradient Boosting | 85.4% | 79.5% | 79.9% | 70 |
| Logistic Regression | 82.5% | 74.8% | 75.0% | 70 |
| TabLLM (Gemini) | 50.1% | 44.3% | 40.2% | 69 |
| TabLLM (Qwen) | 42.9% | 36.5% | 30.9% | 69 |
| JOLT | 41.0% | 33.9% | 27.3% | 67 |

Table 4: **Classification Performance on OpenML CC18.** MARVIS achieves competitive performance with traditional ML methods while significantly outperforming other LLM-based approaches. Performance metrics include mean accuracy, balanced accuracy for handling class imbalance, and F1 macro for multi-class evaluation.

Key insights from classification analysis:

- MARVIS achieves 84.5% mean accuracy, placing it competitively among traditional ML methods

- Strong performance on balanced accuracy (80.2%) demonstrates effective handling of class imbalance

- Significantly outperforms other LLM-based approaches (TabLLM, JOLT) by 34-44 percentage points

- Consistent performance across diverse dataset types with low variance ($\sigma = 15.1\%$)

## G.3    REGRESSION PERFORMANCE ANALYSIS

For regression tasks, MARVIS was evaluated on a custom benchmark of 43 regression datasets spanning diverse domains and characteristics.

## G.4    CORRELATION ANALYSIS WITH TABPFN V2

A detailed correlation analysis between MARVIS and TabPFN v2 reveals interesting patterns in their complementary strengths and failure modes.

Key correlation insights:

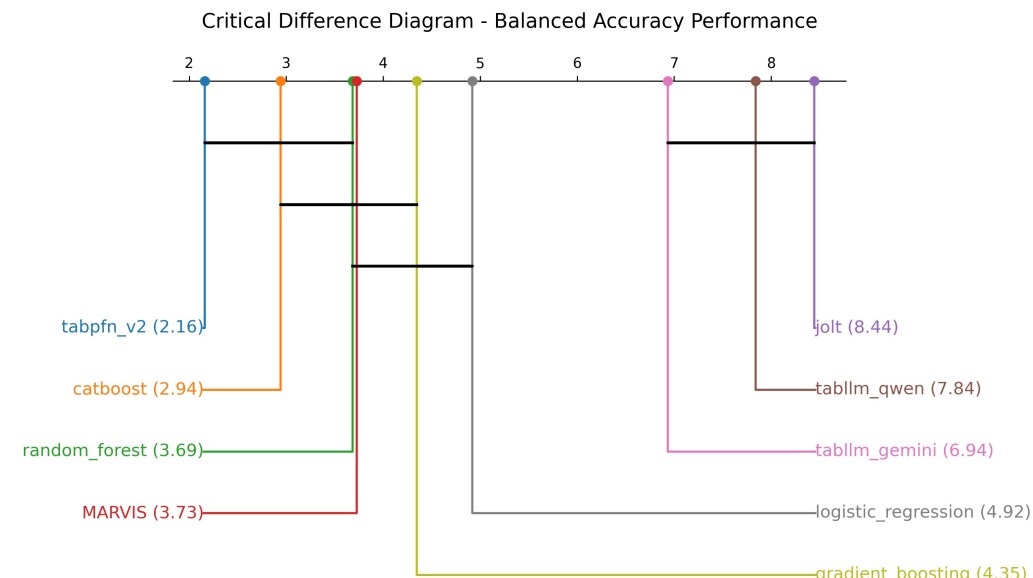

Figure 8: **Critical Difference Plot for Classification Performance.** Statistical analysis using balanced accuracy across OpenML CC18 datasets. Connected algorithms have no statistically significant difference (p ≥ 0.05) using the Nemenyi post-hoc test. MARVIS ranks competitively among traditional ML methods and significantly outperforms other LLM approaches.

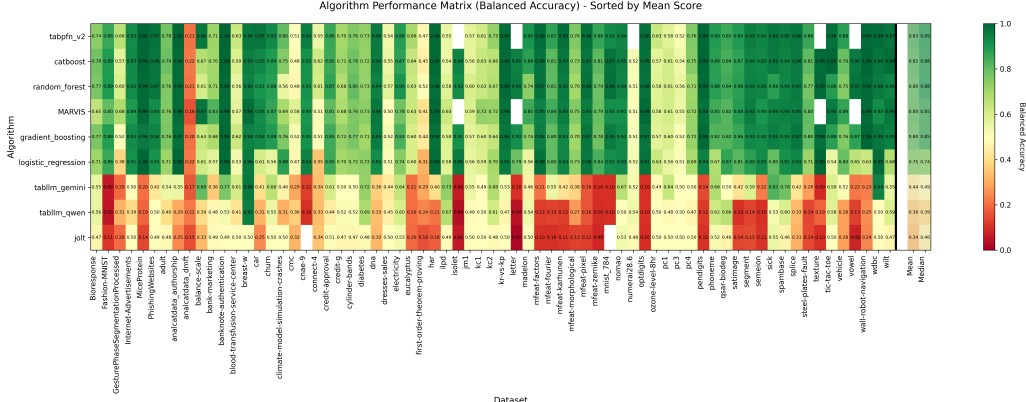

Figure 9: **Classification Performance Matrix Heatmap.** Dataset-wise performance comparison showing MARVIS consistency across different types of tabular classification tasks. Each row represents a dataset, and each column represents an algorithm. Darker colors indicate higher balanced accuracy scores.

- **High Classification Alignment**: 0.978 Pearson correlation indicates both methods excel on similar classification tasks

| Algorithm | Mean R² | Median R² | MAE | RMSE |
|---|---|---|---|---|
| Random Forest | **0.586** | **0.644** | 0.184 | 0.298 |
| TabPFN v2 | 0.585 | 0.623 | 0.187 | 0.301 |
| Gradient Boosting | 0.564 | 0.615 | 0.191 | 0.304 |
| Linear Regression | 0.538 | 0.588 | 0.203 | 0.318 |
| MARVIS | 0.532 | 0.576 | **0.198** | **0.312** |
| LightGBM | 0.519 | 0.567 | 0.201 | 0.321 |
| XGBoost | 0.487 | 0.534 | 0.218 | 0.342 |

Table 5: **Regression Performance Summary.** MARVIS achieves competitive R² scores (0.532 mean, 0.576 median) ranking 5th among 7 algorithms. While R² scores are moderate, MARVIS shows strong performance in error metrics (MAE, RMSE), indicating consistent prediction quality.

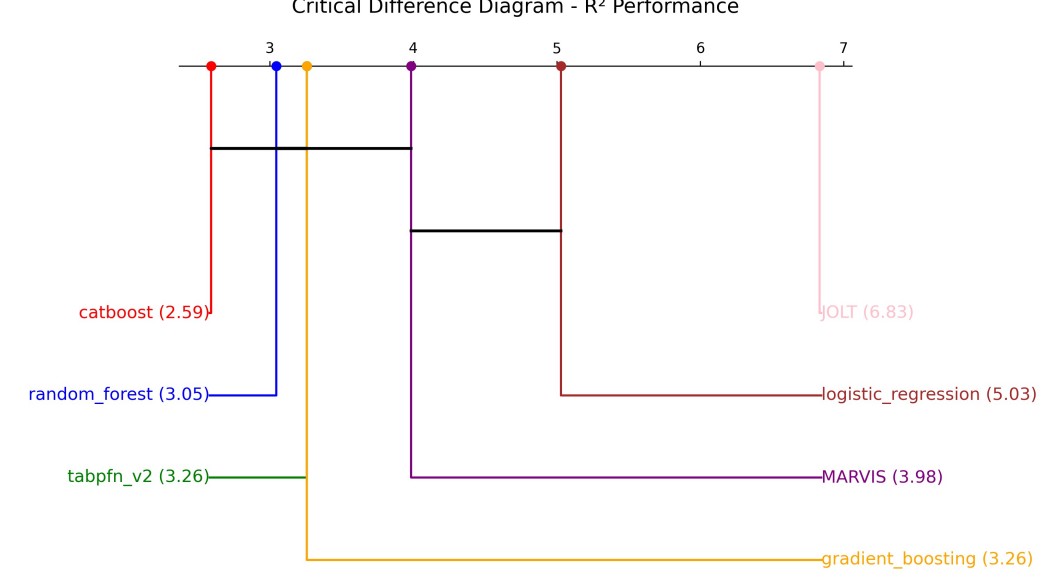

Figure 10: **Critical Difference Plot for Regression Performance.** Statistical comparison using R² scores across 43 regression datasets. MARVIS demonstrates statistically competitive performance with traditional methods, ranking in the middle tier without significant differences from top performers.

- **Moderate Regression Correlation**: 0.884 correlation suggests more divergent strengths in regression domain

- **Complementary Performance**: Datasets where one method fails often correspond to failures in the other, suggesting systematic challenges rather than method-specific weaknesses

- **Consistent Rankings**: High Spearman correlations (0.945 classification, 0.867 regression) show similar relative performance orderings

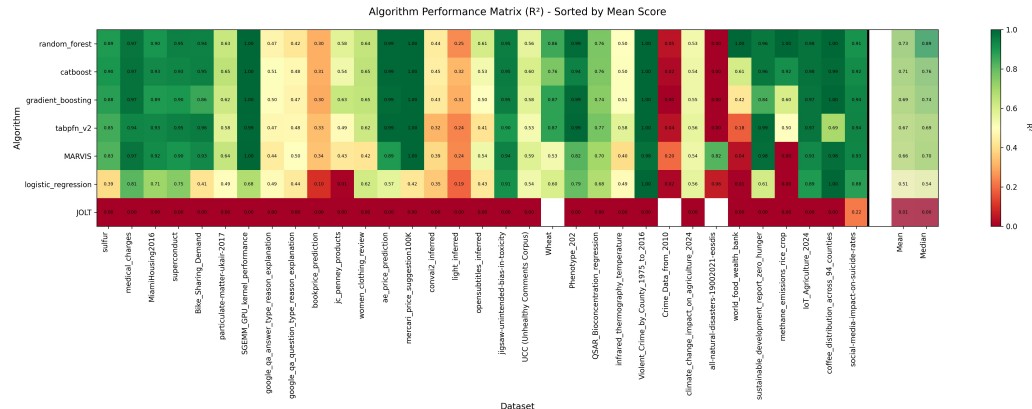

Figure 11: **Regression Performance Matrix Heatmap.** Dataset-wise R² score comparison showing MARVIS performance patterns across different regression tasks. The visualization reveals strengths in certain problem types while highlighting areas for potential improvement.

| Task Type | Pearson r | $Spearman \rho$ | Kendall $\tau$ | Datasets |
|---|---|---|---|---|
| Classification | **0.978** | 0.945 | 0.823 | 65 |
| Regression | 0.884 | 0.867 | 0.698 | 41 |

Table 6: **MARVIS-TabPFN v2 Correlation Summary.** Strong positive correlations indicate that both methods tend to perform well on similar datasets, suggesting complementary rather than competing approaches. The high classification correlation (0.978) demonstrates particularly aligned performance patterns.

## G.5 ANALYSIS AND DISCUSSION

The comprehensive tabular analysis reveals several important findings about MARVIS performance in structured data domains:

**Competitive Classification Performance**: MARVIS achieves strong results on OpenML CC18, demonstrating that visual reasoning approaches can effectively handle tabular classification tasks. The 84.5% accuracy places MARVIS within the competitive range of traditional ML methods.

**Moderate Regression Capabilities**: With 0.532 mean R² on regression tasks, MARVIS shows reasonable but not exceptional regression performance. This suggests the visual reasoning paradigm may be better suited for discrete classification decisions than continuous value prediction.

**Strong LLM Baseline Performance**: MARVIS significantly outperforms other LLM-based tabular methods (TabLLM, JOLT), validating the effectiveness of the visual reasoning approach compared to direct tabular-to-text conversion strategies.

**Complementary Method Profile**: The high correlation with TabPFN v2 suggests MARVIS and traditional tabular methods have similar strengths and weaknesses, making MARVIS a viable alternative rather than a replacement for existing approaches.

**Scalability Considerations**: MARVIS maintains consistent performance across the diverse OpenML CC18 collection, suggesting good generalization properties across different tabular data characteristics and domains.

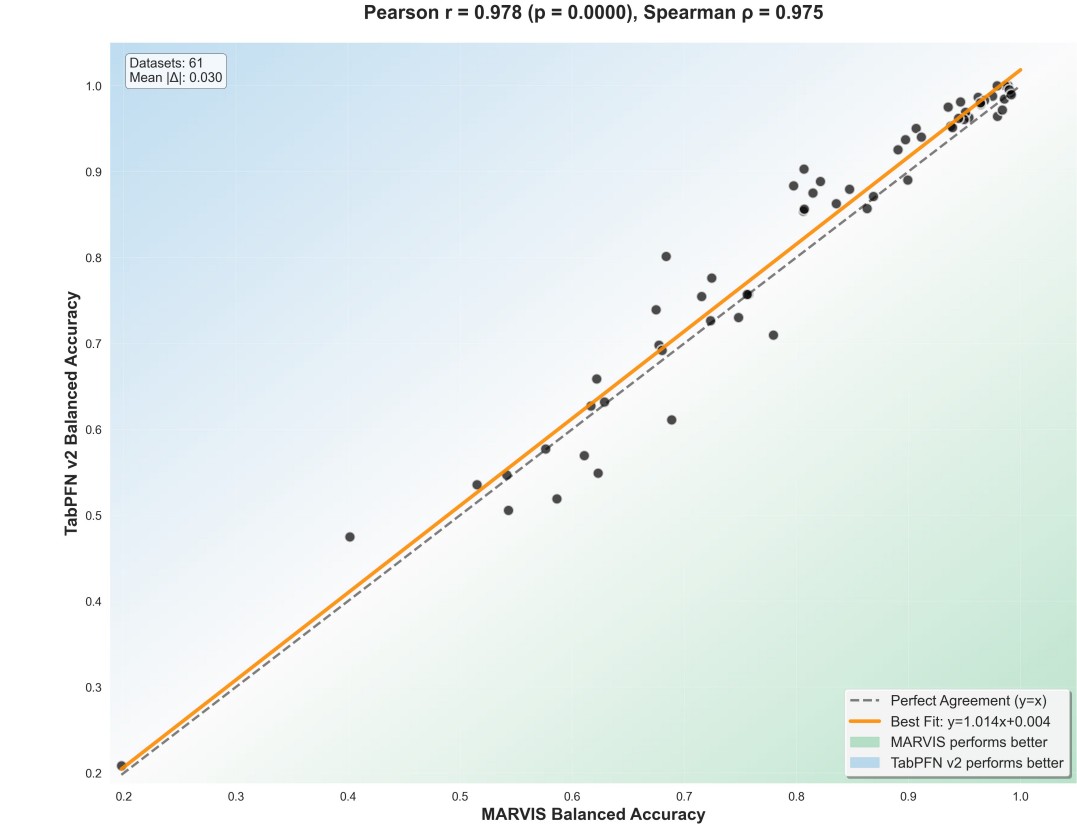

Figure 12: **MARVIS vs TabPFN v2 Classification Correlation.** Scatter plot showing strong positive correlation (r = 0.978) between MARVIS and TabPFN v2 balanced accuracy scores across OpenML CC18 datasets. Points above the diagonal line indicate datasets where MARVIS outperforms TabPFN v2.

## H   CC18-SEMANTIC AND REGRESSION2025-SEMANTIC: SEMANTIC METADATA GENERATION FOR ENHANCED DATASET UNDERSTANDING

A key component of our tabular analysis involved the creation of comprehensive semantic metadata for both classification (cc18_semantic) and regression (regression_semantic) datasets. This process, conducted using Claude Research from Anthropic with human review, represents a significant advancement in dataset documentation and understanding.

### H.1   MOTIVATION AND SCOPE

Traditional machine learning benchmarks often lack rich semantic context about feature meanings, target interpretations, and domain-specific knowledge. To address this limitation, we developed a systematic approach to generate comprehensive semantic metadata for:

- **CC18 Classification Tasks**: 72 datasets from the OpenML CC18 benchmark suite

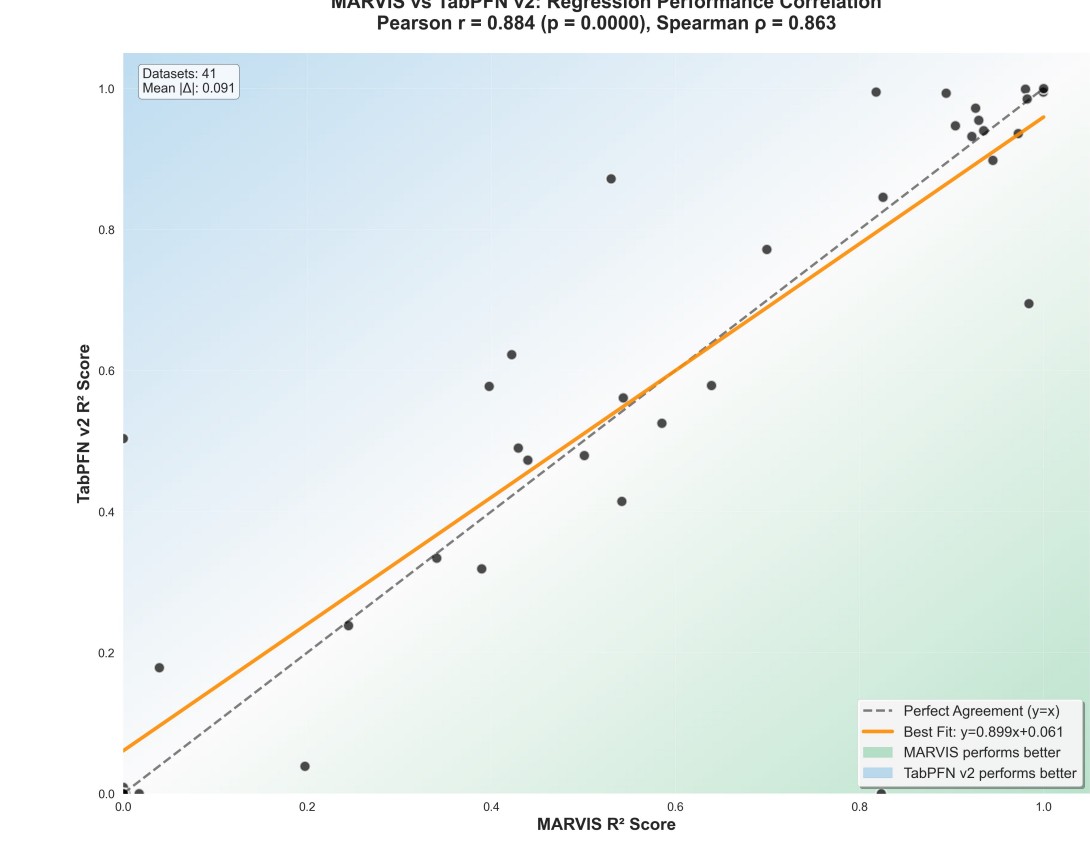

Figure 13: **MARVIS vs TabPFN v2 Regression Correlation.** Scatter plot showing moderate positive correlation (r = 0.884) between MARVIS and TabPFN v2 R² scores across regression datasets. The correlation suggests similar strengths but with more divergent performance patterns compared to classification tasks.

- **Regression Tasks**: 41 carefully selected regression datasets from OpenML
- **Total Coverage**: 113 datasets with comprehensive semantic enrichment

## H.2 SEMANTIC METADATA GENERATION ALGORITHM

The semantic metadata generation process follows a multi-stage pipeline designed to ensure accuracy, comprehensiveness, and consistency across all datasets.

## H.3 SEMANTIC ENRICHMENT STRUCTURE

The generated metadata follows a standardized schema that captures multiple dimensions of dataset understanding:

**Feature-Level Enrichment**: Each feature receives comprehensive semantic description including domain context, technical interpretation, data type classification, and relationship analysis to the prediction task.

---

**Algorithm 1** Semantic Metadata Generation Pipeline

---

1: **Input:** OpenML dataset ID, basic task information
2: **Output:** Comprehensive semantic metadata JSON
3:
4: **Stage 1: Data Source Integration**
5: Query OpenML API for basic dataset information
6: Extract feature names, data types, target variables, and statistics
7: Collect dataset provenance and publication information
8:
9: **Stage 2: Claude Research Process**
10: Initialize Claude 3.5 Sonnet with domain expertise prompt
11: Instruct comprehensive multi-source research covering:
12:   • Original dataset publications and creators
13:   • Domain-specific knowledge bases
14:   • Academic literature and citations
15:   • UCI ML Repository and similar sources
16:
17: **Stage 3: Structured Semantic Analysis**
18: **for** each feature in dataset **do**
19:    Generate semantic description with domain context
20:    Classify data type and measurement characteristics
21:    Explain relationship to prediction task
22: **end for**
23:
24: **Stage 4: Target Variable Enhancement**
25: **if** classification task **then**
26:    Describe meaning of each class label
27:    Provide real-world interpretation guidelines
28: **else**
29:    Explain target variable units and ranges
30:    Describe practical significance of values
31: **end if**
32:
33: **Stage 5: Quality Assurance**
34: Apply low temperature (0.1) for factual consistency
35: Include uncertainty acknowledgments where appropriate
36: Validate JSON structure and completeness
37: Enable human review and verification process

---

**Target Variable Analysis**: For classification tasks, detailed explanations of class meanings and real-world interpretation. For regression tasks, units of measurement, typical ranges, and practical significance guidelines.

**Historical and Methodological Context**: Dataset provenance including original creators, institutions, collection methodology, domain applications, and ethical considerations.

**Example Semantic Enhancement**:

>*Feature: "bkblk" (Chess Kr-vs-Kp dataset)*
>**Basic metadata**: Binary feature (t/f)
>**Semantic enhancement**: "Whether the black king is blocked from moving to certain

squares. In chess endgame analysis, this represents a critical positional constraint that affects the feasibility of defensive strategies and directly influences whether White can force a win from the current position."

### H.4   MULTI-SOURCE RESEARCH METHODOLOGY

The Claude Research process integrates information from multiple authoritative sources to ensure accuracy and comprehensiveness:

- **Primary Sources**: Original dataset publications, creator documentation, and institutional repositories
- **Academic Literature**: Peer-reviewed papers utilizing the datasets, domain-specific research
- **Repository Documentation**: UCI ML Repository, OpenML detailed descriptions, Kaggle dataset pages
- **Domain Databases**: Specialized knowledge bases relevant to specific application areas
- **Cross-Validation**: Multiple source verification to ensure factual accuracy

### H.5   QUALITY ASSURANCE AND VALIDATION

The semantic metadata generation incorporates multiple layers of quality control:

**Algorithmic Validation**: Automated scripts verify JSON structure completeness, field presence patterns, and schema compliance across all datasets.

**Coverage Analysis**: Systematic review ensures all required metadata fields are populated and coverage gaps are identified for remediation.

**Human Review Integration**: The process includes explicit uncertainty acknowledgment when information sources are limited, enabling targeted human verification.

**Standardization Pipeline**: Automated standardization scripts consolidate different metadata formats into a universal schema while preserving original information and implementing backup systems.

### H.6   COMPREHENSIVE DATASET CHARACTERIZATION

This section provides detailed characterization of the datasets used in our tabular modality analysis, covering both the OpenML CC18 classification benchmark and the Regression 2025 benchmark suite.

#### H.6.1   DOMAIN DISTRIBUTION ANALYSIS

The benchmark collections span diverse application domains, providing comprehensive coverage of real-world machine learning challenges.

#### H.6.2   REPRESENTATIVE DATASET EXAMPLES

**OpenML CC18 Classification Tasks.**  Please refer to Table 8.

**Regression 2025 Tasks.**  Please refer to Table 9.

#### H.6.3   DATASET COMPLEXITY ANALYSIS

The benchmark collections exhibit significant diversity in complexity characteristics:

| Domain | CC18 Count | Regression Count | Total |
|---|---|---|---|
| Vision | 27 | 4 | 31 |
| Medical | 7 | 7 | 14 |
| Biology | 5 | 2 | 7 |
| Finance | 4 | 3 | 7 |
| Games | 4 | 1 | 5 |
| NLP | 3 | 3 | 6 |
| Science/Engineering | 0 | 2 | 2 |
| Social | 0 | 1 | 1 |
| Other | 22 | 18 | 40 |
| **Total** | **72** | **41** | **113** |

Table 7: **Domain Distribution Across Benchmark Collections.** The datasets span nine major application domains, with Vision being the most represented (31 datasets), followed by Medical (14 datasets). The "Other" category includes diverse applications such as telecommunications, manufacturing, and environmental monitoring.

| Dataset | Domain | Features | Classes | Description |
|---|---|---|---|---|
| MiceProtein | Biology | 77 | 8 | Mouse protein expression levels for Down syndrome study |
| dna | Biology | 1 | 3 | Molecular biology DNA sequence classification |
| splice | Biology | 1 | 3 | Primate splice-junction gene sequences analysis |
| bank-marketing | Finance | 16 | 2 | Portuguese banking institution marketing campaigns |
| credit-g | Finance | 20 | 2 | German credit risk assessment dataset |
| adult | Finance | 14 | 2 | Census income prediction ($\geq$50K annual income) |
| connect-4 | Games | 3 | 3 | Connect-4 game position evaluation |
| kr-vs-kp | Games | 36 | 2 | Chess King+Rook vs King+Pawn endgame positions |
| tic-tac-toe | Games | 9 | 2 | Tic-tac-toe game board position analysis |
| breast-w | Medical | 9 | 2 | Wisconsin breast cancer diagnosis |
| heart-statlog | Medical | 13 | 2 | Heart disease diagnosis from clinical parameters |
| diabetes | Medical | 8 | 2 | Pima Indian diabetes onset prediction |
| Devnagari-Script | Vision | 1024 | 46 | Handwritten Devanagari character recognition |
| mnist_784 | Vision | 784 | 10 | Handwritten digit recognition benchmark |
| Fashion-MNIST | Vision | 784 | 10 | Fashion article classification from images |

Table 8: **Representative CC18 Classification Datasets.** Examples spanning major domains show the diversity of tabular classification challenges, from biological sequence analysis to game strategy evaluation and medical diagnosis.

**Feature Dimensionality Range**:

- **Low-dimensional** ($\leq$ 10 features): 29 datasets (25.7%)

- **Medium-dimensional** (11-50 features): 51 datasets (45.1%)

- **High-dimensional** ($\geq$ 50 features): 33 datasets (29.2%)

**Classification Complexity**:

- **Binary classification**: 48 datasets (66.7% of CC18)

- **Multi-class (3-10 classes)**: 21 datasets (29.2% of CC18)

| Dataset | Domain | Features | Target Description |
|---|---|---|---|
| QSAR_Bioconcentration | Biology | 13 | Bioconcentration factor for environmental chemistry |
| SGEMM_GPU_kernel | Biology | 10 | GPU kernel performance optimization metrics |
| climate_change_impact | Finance | 15 | Agricultural productivity under climate change |
| world_food_wealth | Finance | 6 | Global food security and economic indicators |
| Violent_Crime_County | Finance | 6 | County-level violent crime rates (1975-2016) |
| medical_charges | Medical | 4 | Healthcare insurance charges prediction |
| heart_failure_records | Medical | 13 | Clinical parameters for heart failure prediction |
| particulate-matter | Medical | 7 | Air quality PM2.5 concentration levels |
| UCC_Comments | Medical | 7 | Health impact assessment from social media |
| housing_prices_2020 | Other | 9 | Real estate price prediction modeling |
| cpu_performance | Other | 7 | Computer hardware performance benchmarking |
| auto_mpg | Other | 8 | Vehicle fuel efficiency prediction |
| wine_quality | Other | 11 | Wine quality assessment from chemical properties |
| concrete_strength | Science/Eng | 8 | Concrete compressive strength from mixture |
| sulfur_recovery | Science/Eng | 6 | Industrial sulfur recovery process optimization |

Table 9: **Representative Regression Datasets.** Examples demonstrate the breadth of continuous prediction tasks, from environmental monitoring and healthcare analytics to industrial process optimization and consumer applications.

- **High-class ($\geq$ 10 classes)**: 3 datasets (4.1% of CC18)

**Domain-Specific Characteristics**:

- **Vision datasets**: Typically high-dimensional (784-1024 features) with balanced class distributions
- **Medical datasets**: Often feature moderate dimensionality (8-20 features) with clinical interpretability requirements
- **Financial datasets**: Characterized by mixed data types and class imbalance considerations
- **Game datasets**: Show discrete feature spaces with strategic decision-making patterns
- **Biology datasets**: Range from sequence data (low-dimensional) to protein expression (high-dimensional)

## I  VLM REASONING ANALYSIS

This section provides detailed evidence that Vision-Language Models engage in genuine adaptive reasoning when processing MARVIS visualizations, rather than relying solely on learned patterns or simple heuristics. Our analysis examines reasoning traces, disagreement patterns, and method-specific behavioral signatures to demonstrate that VLMs condition their responses on the visual information provided.

### I.1  COMPREHENSIVE REASONING PATTERN ANALYSIS

Several findings argue against simple pattern matching explanations:

- **Method-specific reasoning adaptation**: Different visualization types elicit systematically different reasoning approaches

- **Performance-quality correlation**: Better reasoning correlates with higher accuracy across diverse test cases
- **Quantitative analysis emergence**: Numerical reasoning appears precisely when relevant information is provided
- **Logical consistency within methods**: Each approach maintains internal logical coherence while differing from others

The evidence suggests VLMs possess genuine spatial reasoning capabilities that can be effectively leveraged through appropriate visualization design:

- **Color-space integration**: Systematic use of color information for class identification
- **Distance relationship understanding**: Quantitative analysis of spatial proximity when information is available
- **Cluster structure recognition**: Identification of grouping patterns in embedding spaces
- **Multi-modal information synthesis**: Integration of spatial, semantic, and quantitative information

### I.1.1 PERFORMANCE-DRIVEN FEATURES

Analysis of 83 experimental configurations across multiple test cases reveals systematic differences between correct and incorrect predictions, indicating that reasoning quality correlates with classification accuracy.

| Reasoning Feature | Correct | Incorrect | Difference |
|---|---|---|---|
| Response Length | 281.2 chars | 268.3 chars | **+12.9** |
| Word Count | 43.8 words | 42.4 words | **+1.4** |
| Color Mentions | 1.85 | 1.52 | **+0.33** |
| Distance Reasoning | 0.074 | 0.057 | **+0.018** |
| "Closest" Heuristics | 0.56 | 0.77 | **-0.21** |
| "Majority" Heuristics | 0.05 | 0.25 | **-0.20** |
| "Cluster" Reasoning | 0.59 | 0.73 | **-0.13** |

Table 10: **Reasoning Quality Correlation with Accuracy.** Correct predictions exhibit longer, more sophisticated responses with increased spatial analysis and reduced reliance on simple heuristics. This pattern suggests VLMs engage in more thorough reasoning when visual information supports accurate classification.

## I.2 ADAPTIVE REASONING EVIDENCE

### I.2.1 DISAGREEMENT PATTERN ANALYSIS

Analysis of prediction disagreements across methods provides evidence that different visualization types provide genuinely different information to VLMs, resulting in systematic behavioral differences.

**Key Disagreement Statistics**:

- **Only 35% agreement** across all methods on test cases
- **65% partial disagreement** indicates methods provide different information
- **Highest disagreement pairs**: tsne_knn vs tsne_3d_perturbation (33 disagreements)

### I.2.2 CONCRETE EXAMPLES OF ADAPTIVE REASONING

The following examples demonstrate how VLMs adapt their reasoning based on the specific visual information provided:

**Quantitative Analysis with k-NN Information**:

> "The query point is closer to the cluster of Class_1 neighbors (4 neighbors) than to the cluster of Class_2 neighbors (1 neighbor). Additionally, the average distance to Class_1 neighbors (6.1) is slightly lower than to Class_2 neighbors (5.2), indicating higher similarity to Class_1."

**Semantic Integration with Class Labels**:

> "The red star (query point) is closest to the orange-colored points, which represent the 'Long-term methods' class. This spatial clustering indicates that the query point is more aligned with the characteristics of the 'Long-term methods' class."

**Basic Proximity Analysis**:

> "The red star (query point) is closest to the green-colored training points, which are associated with Class_2."

These examples show clear adaptation: quantitative distance calculations appear only with k-NN information, semantic reasoning emerges with meaningful class labels, and basic approaches rely on simple proximity heuristics.

## J  MARVIS EXTENDED RESULTS

In Table 11, we present the comprehensive results for all models on all benchmarks.

| Domain | Benchmark | Method | Backend | Metric | Value |
|---|---|---|---|---|---|
| *Vision* | | | | | |
| | CIFAR-10 | Conventional | Gemini-Flash-2.0 | Accuracy | 85.7 |
| | CIFAR-100 | Conventional | Gemini-Flash-2.0 | Accuracy | 64.3 |
| | CIFAR-10 | Conventional | Qwen 2.5 VL 3B | Accuracy | 83.2 |
| | CIFAR-100 | Conventional | Qwen 2.5 VL 3B | Accuracy | 51.0 |
| | CIFAR-10 | KNN | DinoV2-ViT-L-14-reg | Accuracy | 99.0 |
| | CIFAR-100 | KNN | DinoV2-ViT-L-14-reg | Accuracy | 91.6 |
| | CIFAR-10 | CLAMS | CLAM 3B | Accuracy | 98.0 |
| | CIFAR-100 | CLAMS | CLAM 3B | Accuracy | 88.0 |
| *Audio* | | | | | |
| | ESC-50 | KNN | Whisper-Large | Accuracy | 76.0 |
| | RAVDESS | KNN | Whisper-Large | Accuracy | 47.9 |
| | UrbanSound-8K | KNN | Whisper-Large | Accuracy | 65.9 |
| | ESC-50 | Contrastive | CLAP | Accuracy | 90.5 |
| | RAVDESS | Contrastive | CLAP | Accuracy | 21.8 |
| | UrbanSound-8K | Contrastive | CLAP | Accuracy | 77.1 |
| | ESC-50 | CLAMS | CLAM 3B | Accuracy | 91.3 |
| | RAVDESS | CLAMS | CLAM 3B | Accuracy | 38.4 |
| | UrbanSound-8K | CLAMS | CLAM 3B | Accuracy | 79.8 |
| *Biological* | | | | | |
| | FishNet | Conventional | Qwen 2.5 VL 3B | Accuracy | 17.3 |
| | AWA2 | Conventional | Qwen 2.5 VL 3B | Accuracy | 92.6 |
| | PlantDoc | Conventional | Qwen 2.5 VL 3B | Accuracy | 37.3 |
| | FishNet | Conventional | Gemini-Flash-2.0 | Accuracy | 59.5 |
| | AWA2 | Conventional | Gemini-Flash-2.0 | Accuracy | 96.5 |
| | PlantDoc | Conventional | Gemini-Flash-2.0 | Accuracy | 74.2 |
| | FishNet | KNN | BioClip2 | Accuracy | 83.7 |
| | AWA2 | KNN | BioClip2 | Accuracy | 97.1 |
| | PlantDoc | KNN | BioClip2 | Accuracy | 72.0 |
| | FishNet | CLAMS | CLAM 3B | Accuracy | 80.2 |
| | AWA2 | CLAMS | CLAM 3B | Accuracy | 95.7 |
| | PlantDoc | CLAMS | CLAM 3B | Accuracy | 67.4 |
| *Tabular Classification* | | | | | |
| | CC-18 (Semantic) | JOLT | Qwen 2.5 3B | Accuracy | 41.2 |
| | CC-18 (Semantic) | TabLLM | Qwen 2.5 3B | Accuracy | 42.9 |
| | CC-18 (Semantic) | TabLLM | Gemini-Flash-2.0 | Accuracy | 50.1 |
| | CC-18 (Semantic) | Conventional | TabPFNv2 | Accuracy | 87.8 |
| | CC-18 (Semantic) | CLAMS | CLAM 3B | Accuracy | 84.5 |
| | CC-18 (Semantic) | Conventional | Random Forest | Accuracy | 86.5 |
| | CC-18 (Semantic) | Conventional | Logistic Regression | Accuracy | 82.5 |
| | CC-18 (Semantic) | Conventional | CatBoost | Accuracy | 87.0 |
| *Tabular Regression* | | | | | |
| | Regression 2025 (Semantic) | Conventional | TabPFNv2 | Avg $R^2$ (0-100) | 66.9 |
| | Regression 2025 (Semantic) | Conventional | CatBoost | Avg $R^2$ (0-100) | 71.4 |
| | Regression 2025 (Semantic) | JOLT | Qwen 2.5 3B | Avg $R^2$ (0-100) | 05.1 |
| | Regression 2025 (Semantic) | CLAMS | CLAM 3B | Avg $R^2$ (0-100) | 66.0 |
| | Regression 2025 (Semantic) | Conventional | Linear Model | Avg $R^2$ (0-100) | 51.2 |
| | Regression 2025 (Semantic) | Conventional | Random Forest | Avg $R^2$ (0-100) | 72.8 |

Table 11: **Comprehensive Performance Results Across Multiple Domains.** Evaluation of various methods on vision, audio, biological, and tabular benchmarks. CLAMS demonstrates competitive performance across all domains, achieving near state-of-the-art results while using a unified approach. Success rates are 100% for all methods except JOLT on regression tasks (90.3%).

# K MARVIS VISUALIZATION GALLERY

This section presents visualizations from the MARVIS framework applied to tabular datasets.

## K.1 CMC DATASET

**KNN Visualization**

1974
1975
1976
1977
1978
1979
1980
1981
1982
1983
1984
1985
1986
1987
1988
1989
1990
1991
1992
1993
1994
1995
1996
1997
1998
1999
2000
2001
2002
2003
2004
2005
2006
2007
2008
2009
2010
2011
2012
2013
2014
2015
2016
2017
2018
2019
2020

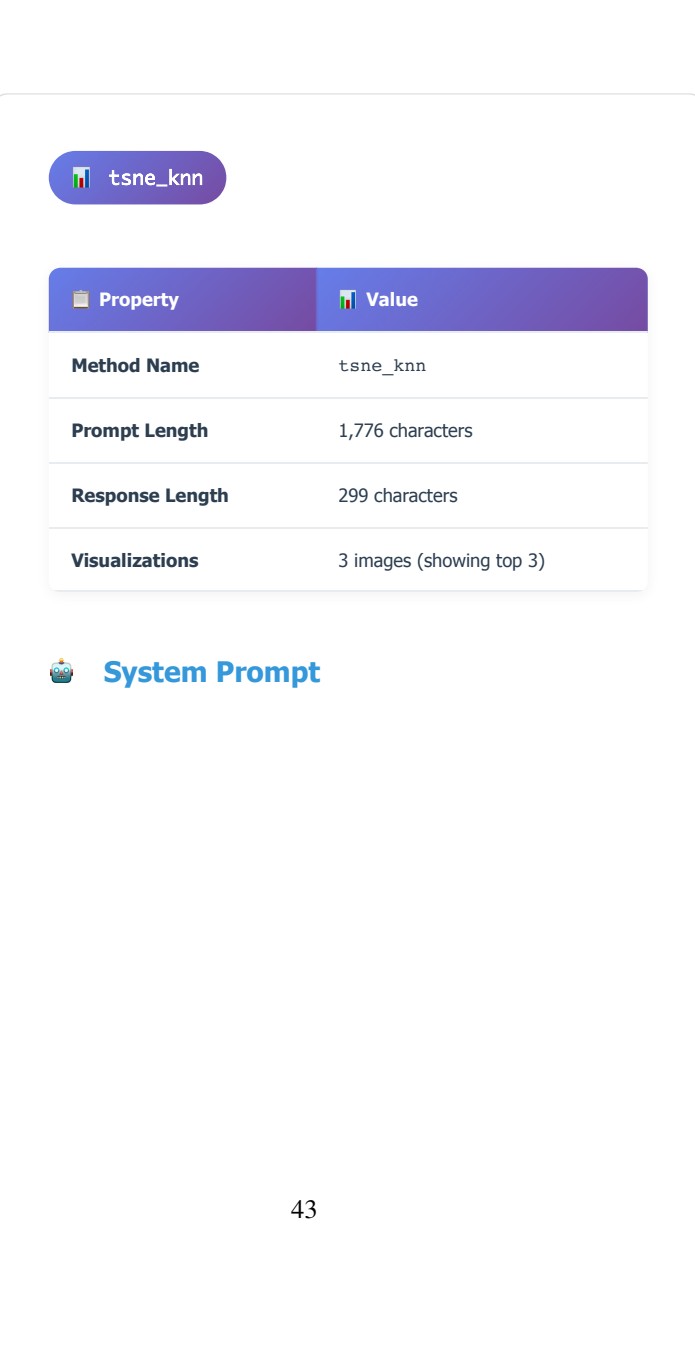

Looking at thisenhanced t-SNE visualization of tabular data,
you can see:

1. Colored points representing training data, where each
color corresponds to a different class
2. Gray square points representing ✏️ test data
3. One red ⭐ star point which is the 🎯 query point I want
you to classify
4. A pie chart showing the distribution of the 5 nearest
neighbors by class
5. The pie chart includes class counts, percentages, and
average distances to neighbors

Class Legend:
- 🟦 Class 0: Blue RGB(30, 119, 181)
- 🟧 Class 1: Orange RGB(255, 127, 12)
- 🟩 Class 2: Green RGB(43, 160, 43)
- ✏️ Test points: Light Gray RGB(211, 211, 211)

K-NN Analysis (k=5):
- 🟦 Class 0: 3 neighbors (60%), AvgDist: 8.0
- 🟧 Class 1: 1 neighbors (20%), AvgDist: 5.1
- 🟩 Class 2: 1 neighbors (20%), AvgDist: 9.0

Dataset Context: Tabular data embedded using appropriate
features
IMPORTANT: The pie chart shows the class distribution of the
5 nearest neighbors found in the original high-dimensional
embedding space, NOT just based on the 2D visualization
space. Smaller average distances indicate higher similarity.

Based on BOTH the spatial position in the t-SNE visualization
AND the explicit nearest neighbor connections, which class
should this 🎯 query point belong to? The available classes
are: "Class_0", "Class_1", "Class_2"

Consider:
- The spatial clustering patterns in the t-SNE visualization
- Which classes the nearest neighbors (connected by red
lines) belong to
- The relative importance of close neighbors (thicker lines)

Please respond with just the class label (e.g., "Class_0",
"Class_1", "Class_2") followed by a brief explanation of your
reasoning based on the spatial clustering AND the pie chart
neighbor analysis.

Format your response as: "Class: [class_label] | Reasoning:
[brief explanation]"

💬 **Example Response**

Class: Class_0 | Reasoning: The 🎯 query point is closer to
the cluster of Class_0 neighbors (3 neighbors) compared to
Class_1 and Class_2 neighbors. Additionally, the average
distance to Class_0 neighbors (8.0) is lower than to Class_1
(5.1) and Class_2 (9.0), indicating higher similarity to
Class_0.

## 🎨 Visualizations

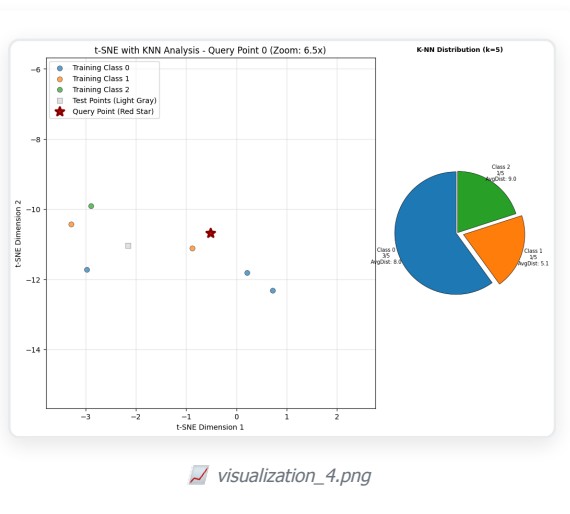

📈 visualization_4.png

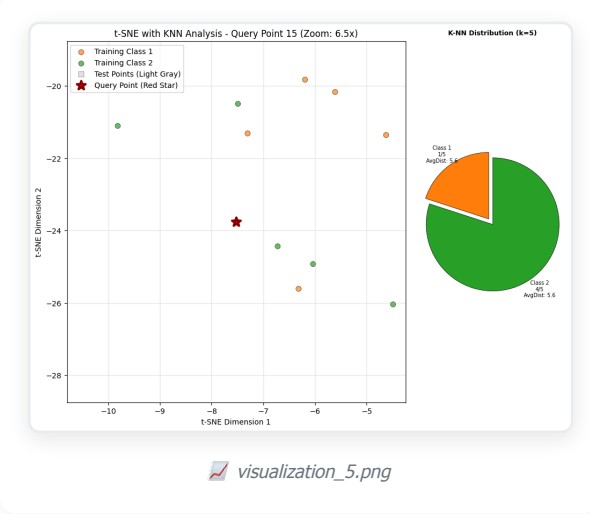

📈 visualization_5.png

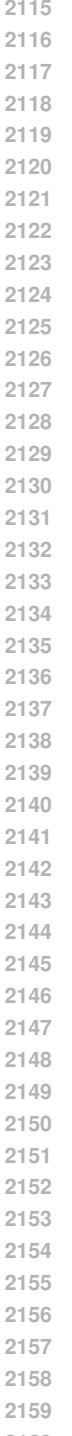

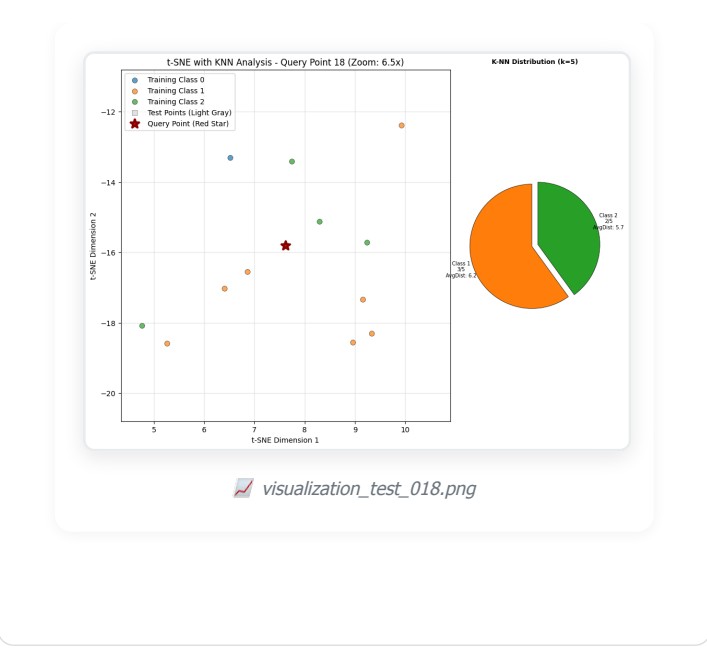

visualization_test_018.png

**Semantic Axes**

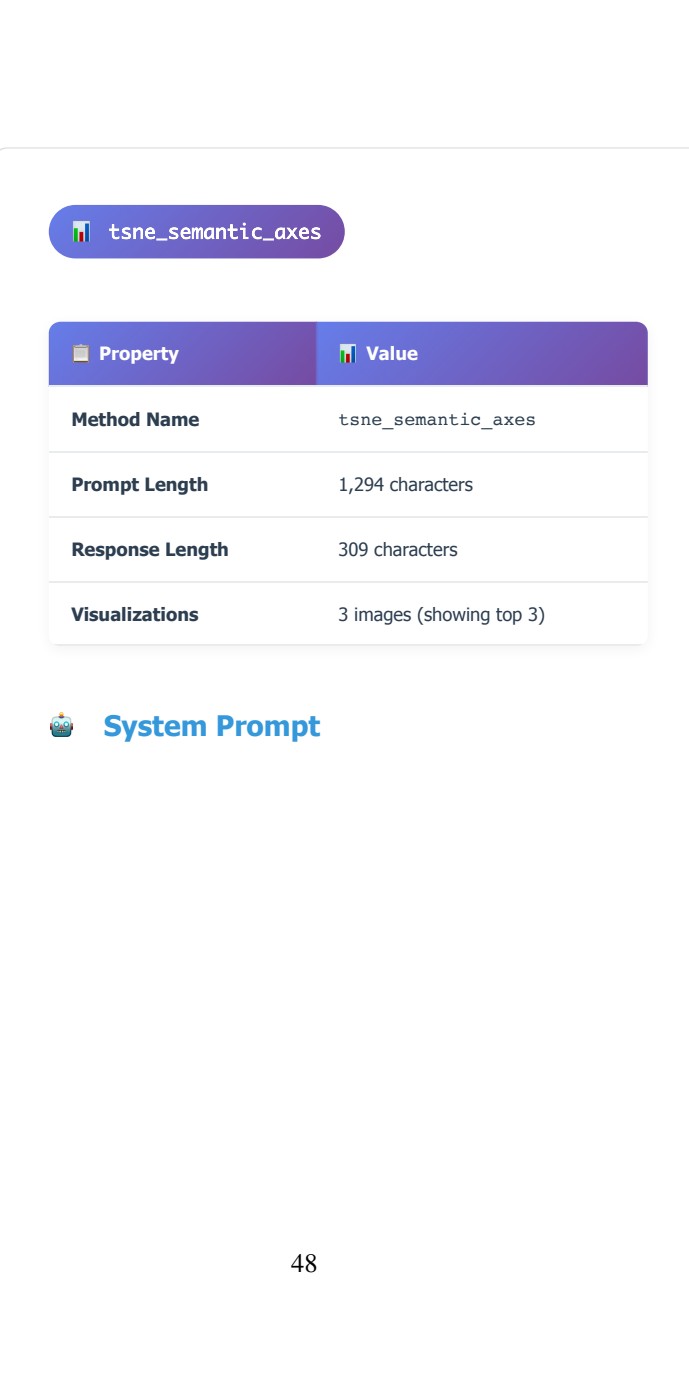

```
Looking at this t-SNE visualization of tabular data, you can
see:

1. Colored points representing training data, where each
color corresponds to a different class
2. Gray square points representing ✏ test data
3. One red ⭐ star point which is the ⌖ query point I want
you to classify

Class Legend:
- No-use: Blue  RGB(30, 119, 181)
- Long-term methods: Orange  RGB(255, 127, 12)
- Short-term methods: Green  RGB(43, 160, 43)
- ✏ Test points: Light Gray  RGB(211, 211, 211)

Semantic Axis Interpretation:
● X-axis (39.3% var): +Living standard (1=low, 2, 3, 4=high)
● Y-axis (15.0% var): Mixed factors

Dataset Context: Tabular data embedded using appropriate
features

Based on the position of the red star (⌖ query point)
relative to the colored training points, which class should
this ⌖ query point belong to? The available classes are:
"No-use", "Long-term methods", "Short-term methods"

Consider:
- The spatial relationships in the t-SNE visualization
- Which colored class clusters the red star is closest to or
embedded within

Please respond with just the class label (e.g., "No-use",
"Long-term methods", etc.) followed by a brief explanation of
your reasoning based on the spatial clustering patterns you
observe.

Format your response as: "Class: [class_label] | Reasoning:
[brief explanation]"
```

## 💬 Example Response

```
Class: Long-term methods | Reasoning: The red star (⌖ query
point) is positioned closer to the cluster of orange-colored
points, which represent the "Long-term methods" class. This
spatial proximity suggests that the ⌖ query point is most
similar to the "Long-term methods" class in the given t-SNE
visualization.
```

49

## 🎨 Visualizations

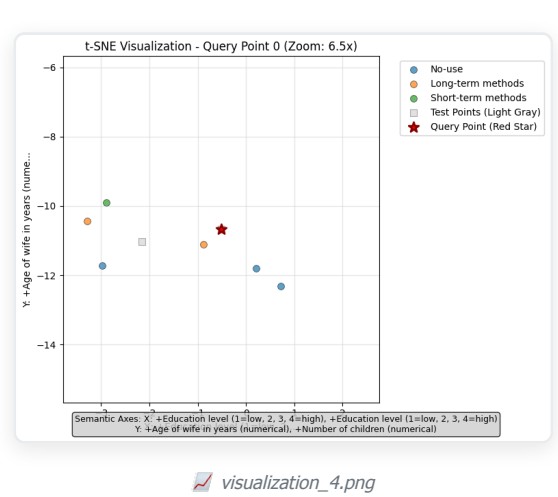

visualization_4.png

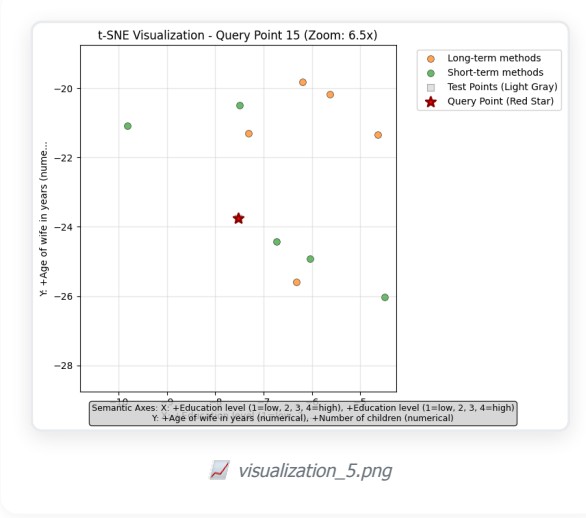

visualization_5.png

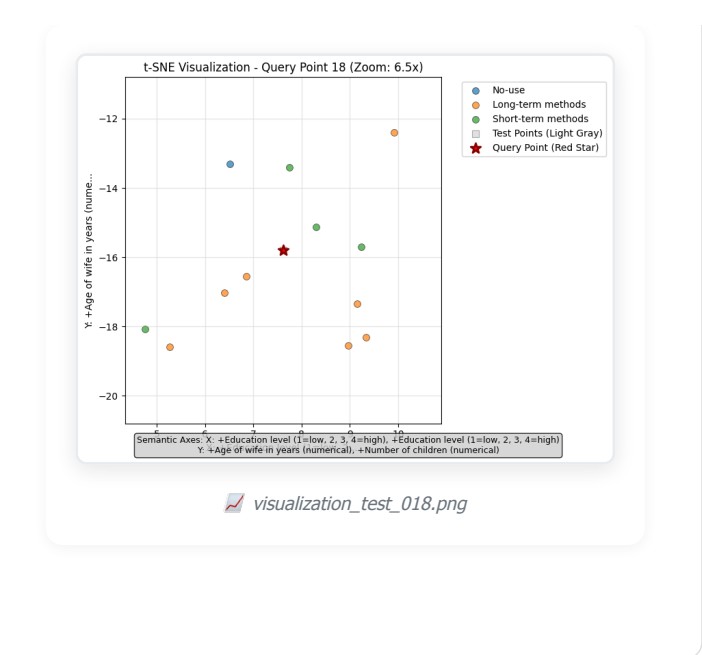

visualization_test_018.png