# OpenReview forum: "MARVIS: Modality Adaptive Reasoning over VISualizations"
_ICLR.cc/2026/Conference — ICLR 2026 Conference Withdrawn Submission_

### Official Review · Reviewer_Cg2V · 2025-10-22

**Soundness:** 2
**Presentation:** 1
**Contribution:** 2
**Rating:** 2
**Confidence:** 4

**Summary:**

This paper is motivated by the observation that although existing LLMs and VLMs demonstrate strong flexibility and generalization capabilities, their performance on specialized tasks still lags far behind that of dedicated predictors. To address this gap, the authors propose MARVIS, a framework that enriches VLMs’ understanding of task-specific contexts by embedding data and enabling the models to analyze various statistical representations—such as t-SNE plots, k-NN graphs, and related forms. In experiments, MARVIS significantly outperforms baseline LLMs and VLMs, achieves performance comparable to specialized predictors, and even surpasses them on certain tasks.

**Strengths:**

1. The motivation is well-founded. Exploring the gap between general intelligence and specialized intelligence is both an interesting and valuable research direction.

2. MARVIS demonstrates strong effectiveness: as shown in Figure 1 and Table 1, VLMs equipped with MARVIS achieve performance comparable to specialized predictors, effectively bridging this gap.

3. The methodology is diverse and thorough. To identify the optimal statistical representation, the authors experimented with 25 different configurations, reflecting substantial exploration and rigor in design.

**Weaknesses:**

1. Some aspects of the background description are not entirely appropriate. To my understanding, the authors’ motivation has two main components: 1) the gap between general and specialized predictors, and 2) the issue of data leakage. The authors emphasize their goal of "combining the reasoning capabilities of LLMs without requiring modality-specific fine-tuning or exposing sensitive data". However, citing data leakage as a justification for avoiding fine-tuning may not be fully convincing, since the tasks explored in this paper mainly involve common, traditional benchmarks. That said, I do recognize the value of exploring zero-shot methods to enhance existing VLMs; I simply believe the authors should provide a stronger and more compelling rationale for their position.

2. The method description is too brief and lacks clarity. In Section 2, the authors present their proposed method, MARVIS, and divide the pipeline into four stages. However, each stage is described only superficially. In the Embedding Generation stage, it remains unclear which embedding models are used for different tasks. In the Dimensionality Reduction stage, what exactly serves as the input to t-SNE — all training data points or only a subset? This section lacks a formal formulation or mathematical definition. In the Visual Reasoning stage, the authors rely solely on Qwen 2.5-VL 3B as the base VLM. Why was this particular model chosen, and why were larger or alternative VLM series not considered? Is MARVIS specifically tailored to Qwen 2.5-VL 3B, or is it intended to generalize across other architectures? Furthermore, besides the t-SNE and k-NN visual inputs, do the VLMs also take the original images as input? Overall, while the pipeline appears simple in structure, its presentation is unclear and potentially confusing due to the lack of detailed methodological explanation.

Overall, the paper reads more like a technical report than an academic paper. Almost every paragraph includes a link to the appendix, suggesting that the main text merely provides a high-level overview while most of the essential content resides in the supplementary materials. Many critical methodological details, such as the operations of t-SNE, k-NN, and perturbation, are barely discussed in the main body. Moreover, the presentation and layout are problematic: several figures with sparse information occupy nearly an entire page, leaving little room for substantive explanation. As a result, readers must constantly refer to the extensive appendix to understand even the basic methodology. It is unclear whether this structure aligns with standard academic publication practices.

**Questions:**

All of my questions fall under the weaknesses section.

---

> ### Author Response · Authors · 2025-11-13
>
> We thank the reviewer for a detailed analysis.
>
> W1. We appreciate the reviewer's interest in the issue of data leakage. As stated in our introduction, MARVIS is designed to preclude leaking P.I.I. or directly leaking data. For these specific kinds of leakage, the choice of benchmark is immaterial, as this avoidance is inherent in the design of MARVIS. The VLM never sees the actual data, nor does it ever see any P.I.I., only visualizations. Please refer to Appendix Sec. K for examples. Therefore, leakage of the specified types is not possible. We hope this clarifies our intention, and we will be happy to add some clarifying language to a future version of the work.
>
> W2. The reviewer's questions, while reasonable, are, as the reviewer notes, generally addressed in the linked and extensive appendices, which is why we worked to provide them. While we empathize with the reviewer's desire to have more analysis of the method in the main body of the paper, a quick survey of other reviews reveals that other reviewers have an equally strong bias in the main paper against small figures that are hard to read, essential results appearing in the appendix, et cetera. We agree with the general point here, however, that the presentation of the work could be improved, and welcome concrete suggestions to this effect, such as making figures more compact, which we will certainly consider.

---

### Official Review · Reviewer_pjNj · 2025-10-27

**Soundness:** 2
**Presentation:** 2
**Contribution:** 2
**Rating:** 2
**Confidence:** 3

**Summary:**

This paper introduces MARVIS, a training-free method that enables VLMs to perform predictive tasks on any data modality. The core idea is to:
1. Use a specialized model to create latent embeddings of the data.
2. Plot these embeddings (and a query point) onto a 2D visualization using t-SNE.
3. Feed this image to a standard VLM, which then reasons visually about the query point's cluster and neighbors to make a prediction.

This approach achieves performance competitive with specialized models while significantly outperforming other LLM baselines. Its key advantages are data privacy and universality.

**Strengths:**

1. The method is training-free yet achieves results close to specialist models across diverse modalities with a single 3B VLM.
2. Solves a key adoption blocker by ensuring the VLM never ingests the sensitive raw data, only an anonymized plot.
3. Requires no domain-specific finetuning, making it far cheaper and easier to deploy than training specialist models.

**Weaknesses:**

1. Performance is strictly capped by the quality of the upstream embedding model. MARVIS cannot outperform its embedding source.
2. The method relies on t-SNE, which is computationally slow and becomes visually illegible on datasets with millions of data points, limiting its use on web-scale data.
3. Introduces a new set of visualization hyperparameters (e.g., t-SNE perplexity) that must be tuned for optimal VLM performance.

**Questions:**

1. How does the method handle tasks with thousands of classes? A t-SNE plot with 1000+ colors seems visually unmanageable for a VLM.
2. For the tsne_knn method, how much performance comes from true spatial reasoning versus just parsing the explicit KNN text/pie chart provided in the image?

---

> ### Author Response · Authors · 2025-11-13
>
> We thank the reviewer for a detailed analysis.
>
> W1. It is true that the optimal evaluation of all input context forms a kind of heuristic "upper bound" on how good MARVIS can be. But since MARVIS matches or exceeds strong baselines, including in some cases (RAVDESS, URBANSOUND8K) the best specialist predictor, we would argue this is more of a limitation to be documented than a weakness of the method.
>
> W2 / Q1. t-SNE finishes in well under a tenth of a second for most datasets in our study; it is faster than almost every specialized baseline and far faster than even the smallest LLM or VLM. It can also be parallelized and run asynchronously from MARVIS itself. If that is still too slow, we provide ablation results for PCA, which is even faster. As described in Sec. 3, we tune t-SNE zoom factor as a hyperparameter; this controls both the number of colors in each plot and the number of points. In this way, we are able to successfully scale MARVIS even to datasets like FishNet, with 450+ classes and almost 100,000 samples.
>
> W3. This is true. We acknowledge that we could have provided more appendix material on the details of how we conducted our HPO (it was a small grid search on 500 samples from each dataset) and will consider improving this in future revisions.
>
> Q2. Please refer to Sec. 3.1 of our paper, "VLMs reason over their input data and condition their behavior based on the context provided", for evidence that different visualization methods elicit systematically different reasoning approaches, providing strong evidence that VLMs adapt their analysis based on the available visual information content, including KNN texts and pie charts.

---

### Official Review · Reviewer_DBkf · 2025-10-29

**Soundness:** 2
**Presentation:** 3
**Contribution:** 2
**Rating:** 4
**Confidence:** 3

**Summary:**

This paper introduces MARVIS, a training-free framework that enables VLMs to perform predictive tasks across a wide range of data modalities. The main idea is to first generate latent embeddings of the data using a specialized model, then create a 2D visualization of this embedding space using dimensionality reduction techniques. A VLM is then prompted to reason over this visualization to make a prediction. The authors demonstrate that this approach achieves performance comparable to specialized models and yields better results than other LLM/VLM-based methods.

**Strengths:**

- The paper recasts diverse modalities as a common visual interface to a VLM, using a simple embed→visualize→infer pipeline that avoids modality‐specific fusion or fine-tuning.
- This work facilitates reproducibility by releasing code and evaluation scripts and by introducing two semantically annotated benchmarks—CC18-Semantic and Regression2025-Semantic—which address the gap in semantic labels for the evaluation of tabular data.

**Weaknesses:**

- Unconvincing example for problem-specific reasoning capabilities
The authors claim that MARVIS enables transparent, interactive reasoning beyond black-box predictions. However, their main evidence, the dialogue in Fig. 4, is internally inconsistent and unconvincing. In that dialogue, the VLM says, “While I cannot directly visualize the data here,” indicating it is not reasoning over the provided image. This contradicts MARVIS’s core basis of image-grounded reasoning. Consequently, its subsequent "insights" are only generic textbook definitions of t-SNE and universally applicable advice, such as "Increase Training Data." These responses could have been generated without any visual input, creating a notable gap between the paper's claims and its evidence. As a result, this claimed benefit of MARVIS remains unsubstantiated, and its practical utility for generating genuine, data-driven insights is questionable.

- Overstated claim on “training-free”
The "training-free" claim is overstated because the method's effectiveness depends on a complex, sensitive "visualization engineering" stage that requires dataset-specific tuning. According to the ablation study in Figure 3, the choice of visualization strategy alone accounts for a range of ~25% to over 50% accuracy. This reveals that the method is not robust to these choices. The burden of optimization is simply shifted from model fine-tuning to a manual, heuristic-driven search over a new set of hyperparameters in the visualization pipeline. These include not only the choice of context strategy (tsne_knn, tsne_semantic_axes, etc.) but also t-SNE parameters and visual settings, such as the "zoom factor."
Moreover, the paper offers no clear principles for selecting these parameters a priori for a new task. This means achieving the reported performance would likely require a costly, trial-and-error process, weakening the "plug-and-play" utility of the proposed framework.

- Lack  of justification for the default experimental setting
The experiments in Table 1 use the tsne_knn configuration, yet the ablation in Figure 3 indicates that other configurations, such as tsne_perturbation_axes, achieve higher accuracy. The justification for this choice is that the configuration "exposes less information... and therefore better reflects real-world use," which is unconvincing and requires further elaboration.

- Inference latency
The paper reports an inference time of 0.5-2.0 seconds per sample. While acceptable for some interactive use cases, this is much slower than a typical specialized model, which performs a single, fast forward pass. This latency could be a bottleneck for applications requiring large-scale batch processing or real-time responses. A more direct comparison of inference speed against the baselines would be beneficial.

Minor Issues
- Line 102: Github -> GitHub
- Line 187: T-SNe -> t-SNE
- Line 319: TSNe -> t-SNE
- In Figure 4, the text within the dialogue bubbles is difficult to read due to its small size and has been truncated with ellipses despite ample empty space.

**Questions:**

- Computational Cost: Could you please elaborate on the inference latency? How does the 0.5-2.0s per sample compare to the specialized model baselines? Are there opportunities to optimize this, for example, by batching the generation of visualizations or the VLM inference step?

- Choice of Visualization Method: The tsne_knn method was used for the main experiments, despite tsne_perturbation_axes and tsne_semantic_axes showing higher mean accuracy in the ablation study. Could you further justify this choice? In particular, could you elaborate on the statement that the chosen method “exposes less information” and “better reflects real-world use”? Is the performance gap between these top configurations statistically significant?

- Robustness to Embedding Quality: The experiments use state-of-the-art embedding models. How does MARVIS perform if a weaker or less suitable embedding model is used for a given modality? Does the performance degrade gracefully, or does the approach fail if the embedding space is not well-structured?

- Baseline Performance on Regression: In the tabular regression results (Table 1), the LLM/VLM baseline (JOLT) has an R2 score of just 5.1, while MARVIS achieves 66.0. This gap is exceptionally large compared to other tasks. Could you confirm this baseline result is correct and, if so, offer any insights into why existing LLM-based tabular methods fail so dramatically on this benchmark while MARVIS succeeds?

---

> ### Author Response · Authors · 2025-11-13
>
> We thank the reviewer for a detailed analysis.
>
> W1. With respect, the reviewer's argument here addresses a claim we do not make. Our actual claim (Contribution 2) is that MARVIS "reasons over the available information sources, implicitly analyzing and balancing them to improve its own predictive power.", supported in 3.1. The raw data is typically not an available information source in MARVIS. In the example we provide, the VLM states (correctly) that it cannot directly access the raw data, but only intermediate artifacts (embedding visualizations). This is evidence that the VLM correctly understands its situation, and the nature of the analysis to be conducted (over the supplied context, which does not include raw data). It is true that, without knowing more about the problem (I.E., exposing details about the data), the VLM cannot provide very sophisticated guidance about how to improve performance; however, this is a weakness that could be addressed in subsequent turns by providing samples of raw data or metadata after the predictions are rendered, should the user wish to do so. We do not propose to recreate here prior work that has demonstrated that LLMs can reason about their input data.
>
> W2. We respectfully argue that the claim is quite precisely and correctly stated. MARVIS is "training-free". We alter no weights and perform no gradient updates. Few, if any, methods are "robust" to suboptimal hyperparameter choices. Whether, for a particular use case, one might prefer training to optimizing hyperparameters for downstream datasets, is not in the scope of a research paper. It is true that we offer no novel HPO frameworks in this work; however, as such frameworks already exist, and as inference is fast and inexpensive compared to training, we believe that this choice is reasonable. We do acknowledge that we could have provided more appendix material on the details of how we conducted our HPO (it was a small grid search on 500 samples from each dataset) and will consider improving this in future revisions.
>
> W3 / Q2. We are not certain why the reviewer finds the reason unconvincing. The two methods which outperformed tsne_knn in our ablation required exposing metadata such as column names to the VLM, and we thought it was more interesting and more useful to show that the system worked almost as well without access to this metadata, since in the real world, practicioners often wish to avoid exposing sensitive data via APIs. We hope that this is informative for the reviewer, and we will try to add a few clarifying sentences to future versions of the work.
>
> W4 / Q1. We agree that a thorough latency comparison would have been informative, and hope to be able to include this in the future. Unfortunately, conducting such a comparison with proper controls across several modalities is simply beyond the scope of what we could accomplish in this work. We did not optimize MARVIS for inference speed; with quantization and batching it could be many times faster, but would still likely be slower than small specialized models.
>
>  Q3. This is a very interesting question which we did not explore! However, we would hypothesize that poorly structured embedding spaces would be significantly less useful for MARVIS (and perhaps in general).
>
> Q4. Sadly, this result is correct. As no public benchmark which compares all of these methods exists, we had to create one. We conducted very extensive tuning of JOLT to try to improve this performance. We believe that JOLT in particular, and conventional LLM/VLM baselines in general, struggle to cope with the long numeric strings and massive within-task variance present in most raw regression data, which, when serialized tends to consist of thousands of tokens like this --
>
> Female,18-20,Asian,23.1,22.7,0.8,0.8875,34.9775,35.2,34.54,34.7875,35.2275,34.9275,35.2275,35.26,35.27,35.1875,35.15,35.305,34.8575,34.9975,35.3425,35.3,34.39,34.185,34.475,34.0425,34.795,35.175,35.045,35.775,35.6675,35.715,36.95
> Female,18-20,White,22.9,22.8,0.8,0.8375,35.99,36.1125,35.505,35.57,36.03,36.0275,35.8275,36.065,36.17,36.0625,36.0325,36.21,35.7525,35.8325,36.21,36.17,34.6575,34.91,35.0025,34.7975,35.0425,35.49,35.2625,36.24,35.9725,36.0275,36.85
>
> MARVIS succeeds because the inputs and the target space are visualized rather than serialized, which simplifies the presentation of the problem dramatically for the VLM.

---

### Official Review · Reviewer_E6Cn · 2025-10-31

**Soundness:** 3
**Presentation:** 1
**Contribution:** 2
**Rating:** 2
**Confidence:** 4

**Summary:**

This paper aims to investigate the reasoning capabilities of large language models (LLMs) and the representational power of predictive models without requiring modality-specific fine-tuning or access to sensitive data. The core idea is to project data from any modality into a meaningful embedding space, where visualization of these embeddings enables vision-language models (VLMs) to perform accurate reasoning without modality-specific training.

**Strengths:**

1. The idea of performing embedding visualization and using it as a shared representation space across different data modalities is interesting.

2. Without additional fine-tuning, the combination of feature visualization and VLM-based reasoning demonstrates strong performance on several predictive tasks across diverse modalities.

**Weaknesses:**

1. Although the proposed solution is clearly presented, the connection between the visualized features and how VLMs perform accurate reasoning based on them is not well established. In addition, it remains unclear whether the visualized features introduce potential out-of-distribution (OOD) issues.

2. The overall solution lacks intuitiveness. While the authors make considerable effort to explain their approach, the rationale behind why it works is not sufficiently analyzed. Further investigation into key components, such as feature selection, would strengthen the paper.

**Questions:**

1. The main contribution of this paper lies in exploring feature visualization combined with VLM-based reasoning for zero-shot prediction. It is strongly recommended to provide a clearer visualization of the extracted features.

2. When feeding the feature visualizations into VLMs for reasoning, are there any out-of-distribution (OOD) issues? How robust is the model to such distributional shifts?

3. Given the central role of feature visualization, it is also suggested to include further analysis on the processing of features extracted from foundation models. For example, how would feature selection or dimensionality reduction affect performance? How robust is the model to different feature processing techniques?

4. The reported performance on multi-modal data is promising; however, it remains unclear whether the comparisons are fair. Additional analysis and justification are needed.

---

> ### Author Response · Authors · 2025-11-13
>
> We thank the reviewer for a detailed analysis.
>
> W1: We hear and understand the reviewer's concern about potential out-of-distribution (OOD) issues. We are not certain what this would mean in this context, however, since the only data the VLM sees in our core experiments are charts and plots, and charts and plots are essentially always "in-distribution" data, from the VLM's perspective. For OOD analyses of our embedding generating models, we would refer the reviewer to the papers which introduced those models.
>
> W2: While we already provide an explanation for why the approach works at a high level in Sec. 2, we acknowledge the reviewer's desire to understand this in greater detail. Regarding visual feature selection, we would refer the reviewer to Fig. 3, which contains the results from 28 unique ablations on the choice of visualized features, and Appendix Sec. E, which contains further detail on each method and why we believe T-SNe+KNN is the most effective (tl;dr, T-SNe is effective at capturing reduced-dimension local relationships in a way that is easy for the VLM to analyze, and the KNN pie chart provides vital secondary context in high-dimensions).
>
>  Q1: We respectfully refer the reviewer to Appendix Sec. K, which contains clear visualizations of the extracted features and the prompt provided to MARVIS at inference time, and hope this addresses the concern.
>
> Q2: Please see W1.
>
> Q3: We would respectfully contend that this is primarily a consideration for the authors of the papers which introduce these foundation models; we do not claim any FM as a contribution, we simply process their embeddings. This does, of course, mean that we rely on these embeddings to be high quality, but in practice, we find that they generally are. Of course there can be exceptions, as we acknowledge in the text.
>
> Q4: We are not certain what aspect of our comparisons strikes the reviewer as potentially unfair, or what analysis and justification would be convincing, and so cannot speak to this point in detail.

---

### Note · Authors · 2025-11-14

**Comment:**

We thank the reviewers for their attention. Because of concerns raised by reviewers, we will withdraw this work from consideration at this time. Although we do not necessarily agree with every point raised, we respect the time and effort expended, and hope for a happy outcome at some future date.

**Withdrawal Confirmation:**

I have read and agree with the venue's withdrawal policy on behalf of myself and my co-authors.